

# Why $CO_2$ cools the middle atmosphere – a consolidating model perspective

Helge F. Goessling[1] and Sebastian Bathiany[2]

[1]Alfred Wegener Institute, Helmholtz Centre for Polar and Marine Research, Bremerhaven, Germany
[2]Wageningen University, Wageningen, Netherlands

*Correspondence to:* Helge F. Goessling (helge.goessling@awi.de)

**Abstract.** Complex models of the atmosphere show that increased carbon dioxide ($CO_2$) concentrations, while warming the surface and troposphere, lead to lower temperatures in the stratosphere and mesosphere. This cooling, which is often referred to as "stratospheric cooling", is evident also in observations and considered to be one of the fingerprints of anthropogenic global warming. Although the responsible mechanisms have been identified, they have mostly been discussed heuristically,

incompletely, or in combination with other effects such as ozone depletion, leaving the subject prone to misconceptions. Here we use a one-dimensional window-grey radiation model of the atmosphere to illustrate the physical essence of the mechanisms by which $CO_2$ cools the stratosphere and mesosphere: (i) the *blocking effect*, associated with a cooling due to the fact that $CO_2$ absorbs radiation at wavelengths where the atmosphere is already relatively opaque, and (ii) the *indirect solar effect*, associated with a cooling in places where an additional (solar) heating term is present (which on Earth is particularly the case

in the upper parts of the ozone layer). By contrast, in the grey model without solar heating within the atmosphere, the cooling aloft is only a transient blocking phenomenon that is completely compensated as the surface attains its warmer equilibrium. Moreover, we quantify the relative contribution of these effects by simulating the response to an abrupt increase in $CO_2$ (and chlorofluorocarbon) concentrations with an atmospheric general circulation model. We find that the two permanent effects contribute roughly equally to the $CO_2$-induced cooling, with the indirect solar effect dominating around the stratopause and

the blocking effect dominating otherwise.

## 1  Introduction

The laws of radiative transfer in the Earth's atmosphere are a key to understand our changing climate. With the absorption spectra of greenhouse gases as one central starting point, climate models of increasing complexity have been built during the last decades. These models show that increased carbon dioxide ($CO_2$) concentrations, while warming the surface and the

troposphere, lead to lower temperatures in the middle atmosphere (MA; the stratosphere and the mesosphere) (Manabe and Strickler, 1964; Manabe and Wetherald, 1967, 1975; Fels et al., 1980; Gillett et al., 2003). Meanwhile, observations show a cooling trend in the MA during the satellite era until the most recent years; the negative trend is especially large in the upper stratosphere and in the mesosphere, although uncertainties also increase with height (Beig et al., 2003; Randel et al., 2009; Liu and Weng, 2009; Beig, 2011; Seidel et al., 2011; Thompson et al., 2012; Huang et al., 2014).



Attribution studies have concluded that the depletion of stratospheric ozone was probably the main driver of the cooling in the lower stratosphere (Ramaswamy and Schwarzkopf, 2002; Shine et al., 2003; Thompson and Solomon, 2005, 2009; Forster et al., 2007; Santer et al., 2012), especially in the Antarctic spring (Ramaswamy et al., 2001; Thompson and Solomon, 2009). In addition, the roles of volcanoes and atmospheric dynamics (Thompson and Solomon, 2009), stratospheric water vapour
(Ramaswamy et al., 2001; Shine et al., 2003; Maycock et al., 2011; Seidel et al., 2011), and climate variability (Seidel et al., 2011) have been discussed. The increase of $CO_2$ concentration contributed to the decrease of lower stratospheric temperatures, but only to a small extent (Shine et al., 2003). In the middle and upper stratosphere (and beyond), the $CO_2$ increase has probably been the most important reason for the temperature decrease (Ramaswamy et al., 2001; Ramaswamy and Schwarzkopf, 2002; Shine et al., 2003; Thompson and Solomon, 2005). As ozone concentrations are expected to recover in the future, it seems
likely that $CO_2$-concentrations will be of growing importance also in the lower stratosphere (Cordero and Forster, 2006).

The isolated effect of $CO_2$ on temperatures in the MA is rarely explained. Probably the most frequent argument found in textbooks (e.g., Pierrehumbert, 2010; Neelin, 2011) is related to the ozone layer where a considerable part of the locally absorbed radiation is short-wave (solar) radiation. Therefore, the temperature around the stratopause exceeds the corresponding temperature in a hypothetical grey atmosphere by far. Because the main absorption bands of $CO_2$ are in the long-wave (LW)
part and not in the solar part of the spectrum, an increase in $CO_2$ leads to increased emission of LW radiation while the rate of solar heating remains unaltered. The excess of emission compared to absorption leads to a cooling. It should be stressed that the argument is related not to the depletion but to the mere presence of ozone. Although this effect is a major contributor to $CO_2$-induced MA cooling (as we confirm below), it fails to explain the strong observed and simulated cooling in the middle and upper mesosphere where the solar heating is weaker. Neither can it explain an important difference between $CO_2$ and other
long-lived greenhouse gases: While methane and nitrous oxide have a much weaker effect on MA temperatures compared to $CO_2$, chlorofluorocarbons (CFCs) even tend to warm the lower stratosphere (neglecting ozone depletion) (Dickinson et al., 1978; Forster and Joshi, 2005).

More complete explanations discern not only between solar and LW radiation, but treat the LW absorption spectra of greenhouse gases in more detail. Ramaswamy et al. (2001) and Seidel et al. (2011) point out that the balance of LW emission and
LW absorption must be considered: Any greenhouse gas emits simply according to its local temperature, but absorbs radiation emitted from certain distances (represented by radiation mean free paths) according to a weighting function that is determined by the absorption spectrum of the gas and the atmospheric composition (see also Goody and Yung, 1989). At the absorption bands of certain CFCs, the radiation mean free path of the atmosphere is large because the bands are located in the spectral window region. These CFCs thus absorb mainly radiation emitted from the warm surface and lower troposphere, but emit with
the low temperatures of the MA. Consequently, increased CFC concentrations impose a LW warming tendency. Ramaswamy et al. (2001) point out that, in contrast, the radiation mean free path in the $15\,\mu m$ band of $CO_2$ is small, implying that the radiation absorbed by $CO_2$ in the MA mainly comes from the cold tropopause region. This explanation raises the question of whether the existence of a tropopause, that is, a minimum in the vertical temperature profile below the MA, is necessary for MA cooling by $CO_2$. We show that this is not the case.





Ramaswamy et al. (2001) address yet another aspect: The MA responds much faster than the rather inert surface-troposphere system. Hence, when greenhouse gases are added to the atmosphere, initially the MA cools because the radiation mean free path of the atmosphere has been reduced and the radiation arriving in the MA now stems from higher, colder levels of the troposphere. After this first phase of MA temperature adjustment, the surface and troposphere gradually warm. The tropospheric warming *per se* leads to increased upward LW radiation, which tends to reduce the overall cooling of the MA (see also Forster et al., 1997).

All these effects, though mentioned in the literature and included in complex climate models, are rarely discussed together. Furthermore, they are usually explained only heuristically and not demonstrated with a conceptual model. The most popular educational model of the greenhouse effect, a global-mean grey atmosphere model with one ground level and a one-layer atmosphere (e.g., Neelin, 2011; Liou, 2002), can not explain $CO_2$-induced MA cooling. Although Thomas and Stamnes (1999) introduce an atmospheric window to their conceptual model, they do not apply it to explain MA cooling. The same is true for more complex conceptual models (e.g., Pollack, 1969a, b; Sagan, 1969; Pujol and North, 2002), which are also not limited to the ingredients needed to explain how $CO_2$ cools the MA.

Our article aims to provide a consolidating model perspective on $CO_2$-induced MA cooling. The first part has an educational emphasis as we demonstrate the physical essence of the mechanisms involved, using simple variants of a vertically continuous global-mean radiation model: In Sect. 2 we derive the grey atmosphere model from the general radiative transfer equation, in close analogy to such models in the educational literature (e.g., Goody and Yung, 1989; Thomas and Stamnes, 1999). The grey atmosphere model features no permanent MA cooling when applied in its pure form. In Sect. 3 we therefore extend the grey model to the simple case of two LW bands of which one is fully transparent, i.e., the window-grey case. The rigorous derivations in Sects. 2 and 3 may be skipped by readers primarily interested in the resulting explanations. In Sect. 4 we explain $CO_2$-induced MA cooling using the window-grey radiation model, complemented with a simple analogy. In Sect. 5 we first discuss what can, and what can not, be learnt about the effect strengths based on the window-grey model, and then provide a quantitative separation of the effects based on simulations with the atmospheric general circulation model ECHAM6. This is followed by a summary and conclusions in Sect. 6. In addition, we show the relation of the vertically continuous model to discrete-layer models in Appendix A and provide a formal response analysis in terms of partial derivatives with respect to the parameters of the window-grey model in Appendix B.

## 2 The grey-atmosphere model

We first consider a vertically continuous grey atmosphere with horizontally homogeneous (global-mean) conditions. The grey atmosphere is transparent for solar radiation and uniformly opaque for LW radiation. Splitting the electromagnetic spectrum into a transparent solar band and an opaque LW band is a common approximation that is naturally suggested by (i) the well separated emission spectra of the Sun and the Earth (Fig. 1a) and (ii) the shape of the absorption spectrum of the Earth's atmosphere (Fig. 1b). The grey model accounts only for radiation while other processes of energy transfer (most importantly convection) are neglected. Greenhouse gases are assumed to be well-mixed in the vertical and any effects from clouds or



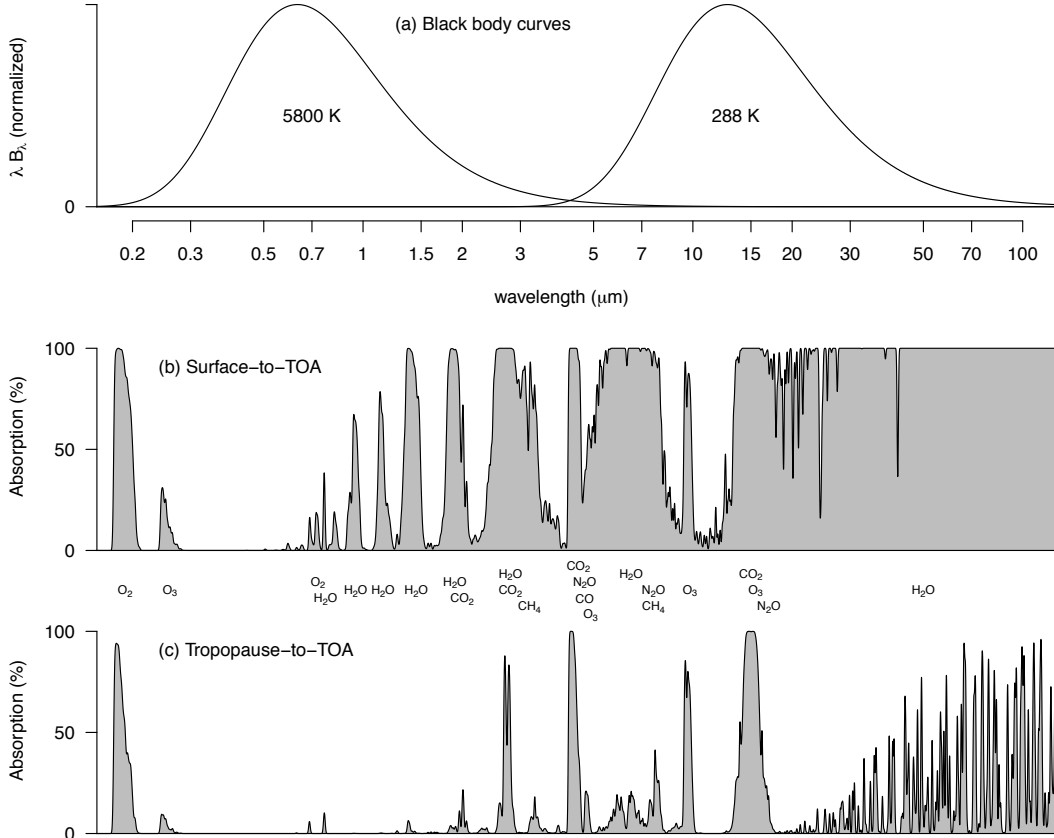

**Figure 1.** (a) Normalized black body curves for 5800 K (the approximate emission temperature of the Sun) and 288 K (the approximate surface temperature of the Earth). (b) Representative absorption spectrum of the Earth's atmosphere for a vertical column from the surface to space. (c) The same but for a vertical column from the tropopause ($\sim$11 km) to space. Spectra based on *HITRAN on the Web*; see Appendix C for details. Figure after Goody and Yung (1989, Fig. 1.1 on p. 4).

aerosols are neglected. Assuming horizontally homogeneous conditions and the absence of scattering we apply the two-stream approximation (Liou, 2002; Pierrehumbert, 2010), meaning that we distinguish only upward and downward propagating radiation (indicated by arrows in the subsequent equations).

In the following we derive the vertical temperature profile of a grey atmosphere, first with only the atmosphere in thermal equilibrium and then assuming equilibrium also for the surface. To this end we start from the differential form of the radiative transfer equation (e.g., Goody and Yung, 1989) with radiance $L$ and source term $J$:

$$\frac{\mathrm{d}L^{\uparrow}(z)}{\mathrm{d}z} = \left[ J - L^{\uparrow}(z) \right] \rho(z)\, k$$
$$= \left[ \sigma\, T(z)^4 - L^{\uparrow}(z) \right] \rho(z)\, k, \tag{1}$$



with geometric height $z$, air density $\rho$, mass absorption coefficient $k$, and Stefan-Boltzmann constant $\sigma$. Using the relative pressure deficit

$$h = 1 - p/p_{srf} \tag{2}$$

as vertical coordinate, where $p$ is pressure and $p_{srf}$ is surface pressure, the radiative transfer equation reads

$$\frac{\mathrm{d}L^{\uparrow}(h)}{\mathrm{d}h} = \left[\sigma T(h)^4 - L^{\uparrow}(h)\right]\alpha. \tag{3}$$

The absorption coefficient $\alpha$ is the only parameter of the grey model and describes the atmospheric opacity in the LW band. Due to our definition of the vertical coordinate $h$, $\alpha$ is independent of $h$ (in fact, it follows from hydrostatic balance that $\alpha = k\,p_{srf}/g$). Also, $h$ is proportional to optical thickness: $\delta = \alpha\,h$. Although we distinguish the parameter $\alpha$ from the vertical coordinate $h$, our model is equivalent to similar approaches in popular textbooks of radiative transfer which usually choose

optical depth ($\tau$, also called optical thickness) as their vertical coordinate. For example, the optical thickness between a height $h$ and the top of the atmosphere, in our case $\alpha(1-h)$, is identical to $\tau_{\infty} - \tau$ in Pierrehumbert (2010), to $\tau$ in Salby (1992), and to $\tau/\overline{\mu}$ in Thomas and Stamnes (1999). In the latter two cases, the vertical axis points downwards, hence the reversed sign. In Thomas and Stamnes (1999), $\overline{\mu}$ appears in the equations as no assumption on the angular distribution of insolation is made in the two-stream approximation.

Equation (3) is known as Schwarzschild's equation and holds analogously for downwelling LW radiation $L^{\downarrow}$. In radiative equilibrium $L$ must be free of divergence because other source/sink terms of heat are neglected. With $L^{\downarrow}_{toa} = 0$, $L^{\downarrow}$ is hence determined by

$$L^{\uparrow}(h) - L^{\downarrow}(h) = L^{\uparrow}_{toa}. \tag{4}$$

Thermal equilibrium for a thin layer of air is given when

$$2\,\epsilon\,\sigma\,T(h)^4 = \epsilon\left(L^{\uparrow}(h) + L^{\downarrow}(h)\right), \tag{5}$$

where $\epsilon = \alpha\,\mathrm{d}h$ is the emissivity, and hence also the absorptivity, of the thin layer for LW radiation. Combining Eqs. (4) and (5) yields

$$\sigma T(h)^4 = L^{\uparrow}(h) - \frac{L^{\uparrow}_{toa}}{2}. \tag{6}$$

Substituting Eq. (6) into the radiative transfer equation (Eq. (3)) gives

$$\frac{\mathrm{d}L^{\uparrow}(h)}{\mathrm{d}h} = -\frac{\alpha}{2}\,L^{\uparrow}_{toa}. \tag{7}$$

Because $\alpha$ is constant, Eq. (7) has the simple solution

$$L^{\uparrow}(h) = L^{\uparrow}_{toa}\left[\frac{\alpha}{2}\,(1-h) + 1\right]. \tag{8}$$





With Eq. (4) it follows further from Eq. (8) that

$$L^{\downarrow}(h) = L^{\uparrow}_{toa} \frac{\alpha}{2} (1 - h).\tag{9}$$

Inserting Eqs. (8) and (9) into Eq. (5) leads to the vertical temperature profile of the equilibrated grey atmosphere:

$$T(h) = \sqrt[4]{\frac{L^{\uparrow}_{toa}}{2\sigma} \left( \alpha (1 - h) + 1 \right)}.\tag{10}$$

Evaluating Eq. (8) at the surface ($h = 0$) gives

$$L^{\uparrow}_{toa} = \frac{L^{\uparrow}_{srf}}{\alpha/2 + 1}.\tag{11}$$

Assuming that the surface is a perfect black body for LW radiation, it is

$$L^{\uparrow}_{srf} = \sigma T^4_{srf}.\tag{12}$$

With Eqs. (11) and (12), the vertical temperature profile described by Eq. (10) can be written as

$$T(h) = T_{srf} \sqrt[4]{\frac{\alpha (1 - h) + 1}{\alpha + 2}}.\tag{13}$$

Equation (13) implies for the near-surface air that

$$T(0) = T_{srf} \sqrt[4]{\frac{\alpha + 1}{\alpha + 2}}.\tag{14}$$

Hence, $T(0) < T_{srf}$. The reason for this discontinuity at the surface is that, in order to attain the same temperature as the surface, the near-surface air would have to receive as much LW radiation from above as it receives from the surface below, which is not

the case (see Eq. (4)). In reality a true discontinuity is prevented by conduction as well as molecular and turbulent diffusion of heat, but sharp temperature gradients right above the surface can still be observed (see for example Pierrehumbert, 2010, whose Eq. (4.45) is identical to our Eq. (14)).

The vertical temperature profile can also be written in terms of the effective radiative temperature of the planet, defined as

$$T_{eff} = \sqrt[4]{\frac{L^{\uparrow}_{toa}}{\sigma}}.\tag{15}$$

Inserting $T_{eff}$ into Eq. (10) gives

$$T(h) = T_{eff} \sqrt[4]{\frac{\alpha}{2} (1 - h) + \frac{1}{2}}.\tag{16}$$

Up to now we have not considered the surface energy balance but determined the vertical temperature profile of an equilibrated grey atmosphere with absorptivity $\alpha$ given an arbitrary surface temperature as lower boundary condition. Equation (13) can thus be interpreted as the *quasi-instantaneous* atmospheric temperature profile. In the following, we consider the situation



where not only the atmosphere but also the surface is in thermal equilibrium. We refer to this situation as the overall equilibrium and denote the corresponding variables with the index $eq$.

Surface equilibrium requires that

$$S = L^{\uparrow}_{srf,eq} - L^{\downarrow}_{srf,eq}, \tag{17}$$

where $S$ is solar radiation absorbed at the surface. Inserting Eq. (4) into Eq. (17) gives

$$S = L^{\uparrow}_{toa,eq}, \tag{18}$$

which is the overall equilibrium condition at the top of the atmosphere. It follows with Eq. (15) that

$$T_{eff,eq} = \sqrt[4]{\frac{S}{\sigma}}. \tag{19}$$

In overall equilibrium, the vertical temperature profile (Eq. (16)) hence becomes

$$T_{eq}(h) = T_{eff,eq} \sqrt[4]{\frac{\alpha(1-h)+1}{2}}. \tag{20}$$

Apart from the different choices of the vertical coordinate, the temperature profile given by Eq. (20) is identical to Eq. (12.21) in Thomas and Stamnes (1999), to Eq. (4.42) in Pierrehumbert (2010), and to Eq. (3.47) in Salby (1992).

Inserting $h = 1$ into our Eq. (20) yields the temperature at the TOA:

$$T_{toa,eq} = T_{eff,eq} \sqrt[4]{\frac{1}{2}}. \tag{21}$$

Finally, combining Eqs. (20) and (14) gives the corresponding equilibrium surface temperature:

$$T_{srf,eq} = T_{eff,eq} \sqrt[4]{\frac{\alpha+2}{2}}. \tag{22}$$

Equations (20)–(22) reveal that, in overall thermal equilibrium, an increase in absorptivity of a grey atmosphere without non-LW heat sources leads to a temperature increase everywhere except at the TOA where the temperature (the skin temperature) is independent of $\alpha$.

Figure 2 shows solutions of Eq. (20) for different values of $\alpha$. In the limit of an almost completely transparent atmosphere ($\alpha \to 0$), the whole atmosphere attains the equilibrium skin temperature (Eq. (21)) and the surface temperature attains the effective equilibrium radiative temperature of the planet (Eq. (19)). Note that the vertically continuous model derived here can be interpreted as a generalization of a discrete-layer model (see Appendix A).

## 3   The window-grey atmosphere model

In reality, the atmosphere is not uniformly opaque for LW radiation, as within the grey approximation, but interacts differently with LW radiation of different wavelengths. To account for this in the simplest possible way, we extend the grey model





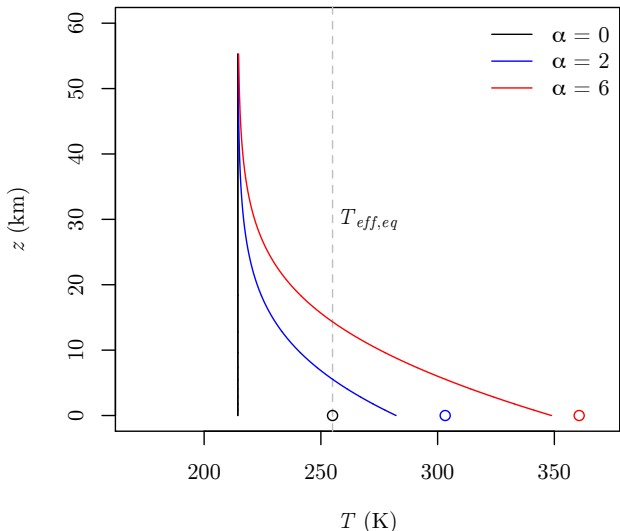

**Figure 2.** Vertical temperature profiles of a grey atmosphere in overall equilibrium for different absorptivities $\alpha$. The latter corresponds to $\alpha_o$ in the window-grey model with $\beta_w = 0$. $T_{eff,eq} = 255\,\mathrm{K}$. The circles at $z = 0$ denote the corresponding surface temperatures. Note that the vertical coordinate $z$ is only approximate height, calculated from $h$ with a constant scale height $H = 8\,\mathrm{km}$ such that $h = 1 - e^{-z/H}$.

(Sect. 2) by splitting the total LW radiation $L$ into two separate LW bands: an opaque band $L_1 = O$ with opacity $\alpha_o > 0$, and a completely transparent (window) band $L_2 = W$ with opacity $\alpha_w = 0$. With $\beta_w = 1 - \beta_o$ describing the fraction of LW radiation from the surface which is directly emitted to space, the resulting window-grey model has only two parameters: $\alpha_o$ and $\beta_w$. Thereby $\beta_w$ is identical to the so-called transparency factor $\mathcal{G}$ in Thomas and Stamnes (1999), but independent of

5    temperature in our case as we neglect Wien's law. This approach represents the so-called window-grey or one-band Oobleck case of a multiband model (Sagan, 1969; Pierrehumbert, 2010). In contrast to the grey-atmosphere model, the window-grey model allows for the existence of a spectral window, like the one the Earth's atmosphere features roughly between $8\,\mu\mathrm{m}$ and $12\,\mu\mathrm{m}$ (Fig. 1b). The window-grey model is depicted in Fig. 3.

With the described simplifications, only radiation in the opaque LW band needs to be considered within the atmosphere, and

10   Eqs. (??)–(??) become analytically solvable. The resulting radiative transfer equation of the window-grey model reads

$$\frac{\mathrm{d}O^{\uparrow}(h)}{\mathrm{d}h} = \left[\sigma T(h)^4(1 - \beta_w) - O^{\uparrow}(h)\right]\alpha_o, \tag{23}$$

the energy balance equation reads

$$2\epsilon_o\sigma T(h)^4(1 - \beta_w) = \epsilon_o\left(O^{\uparrow}(h) + O^{\downarrow}(h)\right), \tag{24}$$

and the surface emission in the opaque band is

15   $$O^{\uparrow}_{srf} = (1 - \beta_w)\sigma T^4_{srf}. \tag{25}$$





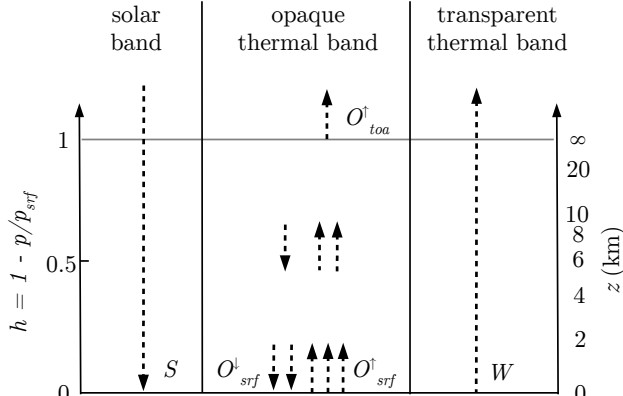

**Figure 3.** Sketch of the window-grey atmosphere model. $S$: solar radiation absorbed at the surface. $O$: radiation in the opaque LW band ($\uparrow$: upwelling, $\downarrow$: downwelling, *srf*: surface, *toa*: top of the atmosphere). $W$: radiation emitted from the surface in the transparent LW band (atmospheric window). $p$: pressure. Interpreting the number of arrows in the opaque LW band as proportional to the radiative flux, the illustrated case corresponds to an equilibrated atmosphere with $\alpha_o = 4$.

Equations (23)–(25) can be solved analogously to the corresponding equations describing the grey case (Eqs. (3), (5), and (12)). This leads to the quasi-instantaneous atmospheric temperature profile of the window-grey model. It is

$$T(h) = T_{srf} \sqrt[4]{\frac{\alpha_o\,(1-h)+1}{\alpha_o+2}}\,. \tag{26}$$

Comparison with Eq. (13) reveals that, with the same surface temperature prescribed as lower boundary condition, the vertical temperature profiles in the grey and in the window-grey case are identical for $\alpha_o = \alpha$; the factor $(1-\beta_w)$ in Eqs. (23)–(25) has cancelled.

To determine the overall equilibrium state, the surface energy balance needs to be incorporated. In overall equilibrium it is

$$S + O^{\downarrow}_{srf,eq} = O^{\uparrow}_{srf,eq} + W_{eq} \tag{27}$$

where

$$W_{eq} = \beta_w\,\sigma\,T^4_{srf,eq}\,. \tag{28}$$

Here we omitted the indices denoting the orientation and vertical position of $W$ (as we already did for $S$) because the only radiation in the window band is the one emitted upward from the surface, and $W$ remains unchanged throughout the atmosphere because $\alpha_w = 0$.

With derivations analogous to the grey case (Sect. 2), one arrives at simple expressions for the overall equilibrium state. The surface temperature for the window-grey model in overall equilibrium is

$$T_{srf,eq} = T_{eff,eq} \sqrt[4]{\frac{\alpha_o+2}{\alpha_o\beta_w+2}}\,. \tag{29}$$





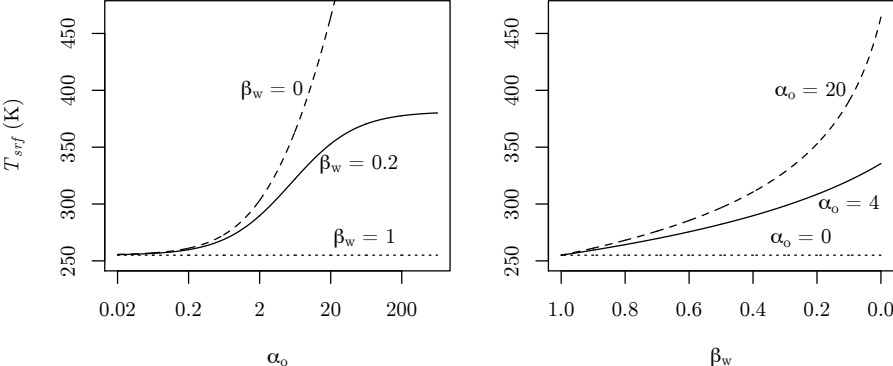

**Figure 4.** The dependence of the overall equilibrium surface temperature on the parameters $\alpha_o$ and $\beta_w$ in the window-grey model (Eq. (29)). $\beta_w = 0$ corresponds to the grey case. $T_{\mathit{eff,eq}} = 255\,\mathrm{K}$.

The corresponding vertical temperature profile reads

$$T_{eq}(h) = T_{\mathit{eff,eq}} \sqrt[4]{\frac{\alpha_o \left(1 - h\right) + 1}{\alpha_o \beta_w + 2}}, \tag{30}$$

which implies for the skin temperature

$$T_{\mathit{toa,eq}} = T_{\mathit{eff,eq}} \sqrt[4]{\frac{1}{\alpha_o \beta_w + 2}}. \tag{31}$$

5    Obviously, with $\beta_w = 0$ Eqs. (29)–(31) are reduced to the grey case (compare Eqs. (20)–(22)).

Equation (29) implies that an increased absorber amount leads to an increased equilibrium surface temperature (Fig. 4), independent of whether the added molecules absorb in the already opaque part of the LW spectrum (increasing $\alpha_o$) or in the window region (decreasing $\beta_w$, that is, "closing the atmospheric window"). In the following section, we discuss the sensitivity of atmospheric temperatures to the model parameters.

## 10    4    The mechanisms of $CO_2$-induced middle-atmosphere cooling

With the window-grey radiation model we are now equipped to investigate the physical essence of $CO_2$-induced MA cooling. In the window-grey model, the response of temperature to changes in the parameters can be quantified with partial derivatives. The different effects of $CO_2$-induced MA cooling can thereby be separated in a formal way. We present such an approach in Appendix B, but constrain the discussion in the following main text largely to the undifferentiated equations.

### 15    4.1    The blocking effect

We first investigate the transient situation in which the assumption of thermal equilibrium is kept for the atmosphere but dropped for the surface. This is a reasonable assumption because the atmosphere adjusts quickly to a change in composition



(on the order of months) while the response of the ocean-dominated surface is very slow (including multi-centennial time scales). The temperature profile for the quasi-instantaneous response is given by Eq. (26) which is valid even if the surface is not in thermal equilibrium. Fig. 5 shows the vertical temperature profile before (blue curve) and after (orange curve) increasing $\alpha$ in the grey case (for which $\alpha$ corresponds to $\alpha_o$ with $\beta_w = 0$).

Inserting $h = 1$ in Eq. (26) we arrive at the corresponding quasi-instantaneous skin temperature:

$$T_{toa} = T_{srf} \sqrt[4]{\frac{1}{\alpha_o + 2}}. \tag{32}$$

Equation (32) implies a cooling at the TOA for increased $\alpha_o$. Furthermore, Eq. (26) implies that at a certain height $\hat{h}_{fast}$ the sign of the fast temperature response due to added greenhouse gases reverses. It is

$$\hat{h}_{fast} = \frac{1}{2}. \tag{33}$$

This implies that at first the upper half of the atmosphere is cooled while the lower half is warmed.

Both the upper-level cooling and the lower-level warming are due to enhanced blocking, that is, a reduced mean free path of LW radiation in response to increased absorptivity. In equilibrium, the emission, determined by the local temperature, and the absorption of radiation are locally balanced. In the upper atmosphere, where downwelling radiation is subordinate, the upwelling radiation received from below comes from higher (and thus colder) levels when the absorptivity of the atmosphere

is increased. Consequently, the air cools until emission and absorption are in balance again. In contrast, in lower levels near the ground, where most of the absorbed upwelling radiation comes directly from the surface (with a fixed temperature), the increased absorptivity mostly affects the downwelling radiation which now comes from lower (and thus warmer) levels, resulting in warming.

As long as the emission from the surface, determined by its temperature, remains unchanged, the surface energy budget is

imbalanced due to the increased downwelling radiation. The surface will thus warm – which is the common greenhouse effect – until a new overall equilibrium is attained. During the gradual ascent of the surface temperature, accompanied by increasing upwelling radiation, the whole atmosphere warms (Eq. 26), and the height at which the sign of the temperature change reverses is shifted upwards. In the grey case, where $\beta_w = 0$, this shift proceeds until the whole atmosphere except the TOA is warmer than originally (red curve in Fig. 5). The upper-level cooling in response to increased absorptivity in a grey atmosphere (and

without absorption of solar radiation within the atmosphere) is thus only a transient effect that vanishes when the new overall equilibrium is reached.

This changes in the presence of an atmospheric window where a part of the surface radiation is directly emitted to space, bypassing the atmosphere. An atmospheric window implies a reduced sensitivity of the surface temperature to the state of the atmosphere (its absorptivity in the opaque band and the corresponding temperature profile, see Eq. (29)) because the

radiation in the opaque LW band becomes less important in the surface energy budget (Eq. (27)) with increasing window size. Consequently, in the presence of an atmospheric window, a permanent cooling at the TOA remains after the surface has equilibrated (Eq. (31)).





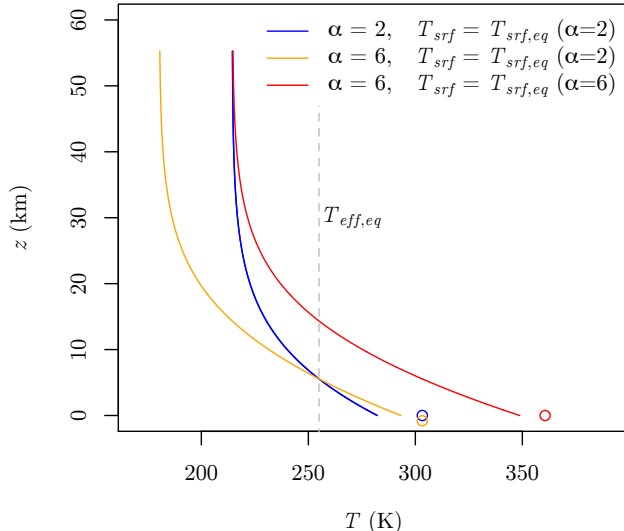

**Figure 5.** Vertical temperature profiles of a grey atmosphere for two equilibrium states and one transient state. While the blue and red curves show the same equilibria as the corresponding curves in Fig. 2, the orange curve shows the transient state that occurs after switching from $\alpha = 2$ to $\alpha = 6$, directly after equilibration of the atmosphere but with $T_{srf}$ still unchanged. Again $\alpha$ corresponds to $\alpha_o$ in the window-grey model with $\beta_w = 0$. $T_{eff} = 255\,\mathrm{K}$. The circles at $z = 0$ denote the corresponding surface temperatures. Note that the vertical coordinate $z$ is only approximate height, calculated from $h$ with a constant scale height $H = 8\,\mathrm{km}$ such that $h = 1 - e^{-z/H}$.

It becomes evident from Eq. (31) that, in contrast to the surface, at the TOA the sign of the temperature response depends on the spectral property of the added absorbers: if they absorb in the already opaque part of the LW spectrum (increasing $\alpha_o$), $T_{toa,eq}$ is decreased (MA cooling), but if they absorb in the transparent part of the LW spectrum (decreasing $\beta_w$, that is, "closing the atmospheric window"), $T_{toa,eq}$ is increased (MA warming) (Fig. 6).

In fact, decreasing $\beta_w$ leads in overall equilibrium to a temperature increase at every height in the atmosphere (Fig. 7, top). In contrast, if molecules absorbing in the opaque LW band are added, the sign of the equilibrium temperature response reverses at a certain height $\hat{h}_{eq}$, with cooling above and warming below (see Eq. (B6); Fig. 7 bottom):

$$\hat{h}_{eq}(\beta_w) = 1 - \frac{\beta_w}{2}. \tag{34}$$

For $\beta_w = 0$, that is in the grey case, $\hat{h}_{eq}$ becomes 1 (the corresponding geometric height $\hat{z}_{eq}$ becomes $\infty$), meaning that no cooling takes place.

We term the permanent cooling in the upper parts of the atmosphere revealed by Eqs. (30) and (31) the *blocking effect* of $CO_2$-induced MA cooling. This presupposes that the main consequence of adding $CO_2$ to the atmosphere is, in terms of the window-grey model, an increase of $\alpha_o$ rather than a decrease of $\beta_w$.

The blocking effect can be understood in terms of the interplay between the sensitivity of the surface temperature to greenhouse-gases on the one hand and the blocking of upwelling LW radiation by greenhouse gases on the other hand: while an

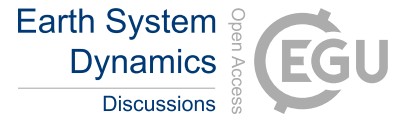

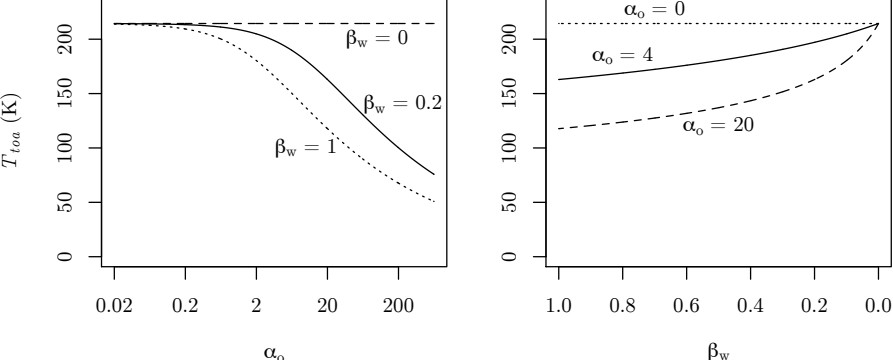

**Figure 6.** The dependence of the overall equilibrium temperature at the top of the atmosphere (the skin temperature) on the parameters $\alpha_o$ and $\beta_w$ in the window-grey model (Eq. (31)). $\beta_w = 0$ corresponds to the grey case. $T_{eff,eq} = 255\,\text{K}$.

atmospheric window diminishes the sensitivity of the surface temperature to $\alpha_o$ (see Eq. (29)), the blocking associated with $\alpha_o$ is independent of the presence or width of an atmospheric window (see Eq. (26)). Only in the grey case, where the sensitivity of the surface temperature is at its maximum (Eq. (29)), the surface temperature response is strong enough to compensate for the blocking effect, resulting in an $\alpha_o$-independent equilibrium skin temperature (compare Eq. (31)).

5      Another way of looking at the permanent blocking effect goes via the emission spectrum of the planet viewed from space (i.e., the upwelling LW radiation at the TOA). If the surface warms in response to an increased $\alpha_o$, the radiation in the window region of the spectrum $W$ will be accordingly stronger, corresponding to a Planck curve at the increased surface temperature. In overall equilibrium the radiation in the opaque band $O_{toa}^{\uparrow}$ must be shifted to lower intensity to compensate for $W$, given that the solar energy input in unchanged. The temperature at the TOA must thus be lower because, as follows from Eq. (24),

$$10 \quad T_{toa} = \sqrt[4]{\frac{O_{toa}^{\uparrow}}{2\sigma(1 - \beta_w)}}. \tag{35}$$

The same argument reveals why the TOA cooling due to enhanced blocking in a grey atmosphere can only be a transient phenomenon: If no window exists, $O_{toa}^{\uparrow}$ must attain its original intensity after equilibration to balance the unchanged solar energy input.

## 4.2   An analogy for the blocking effect

15 While the above explanations are based on mathematical formalism, the following analogy may facilitate an intuitive understanding for the blocking effect of MA cooling by $CO_2$. Consider a building that is heated at a constant rate from inside. In steady state there is a higher temperature inside the building compared to the fixed exterior temperature. The walls of the building represent an analogy to the Earth's atmosphere, with the outer surface as the top of the atmosphere and the inner surface as the atmosphere close to the Earth's surface. The temperature at the outer wall surface is higher than the exterior temperature




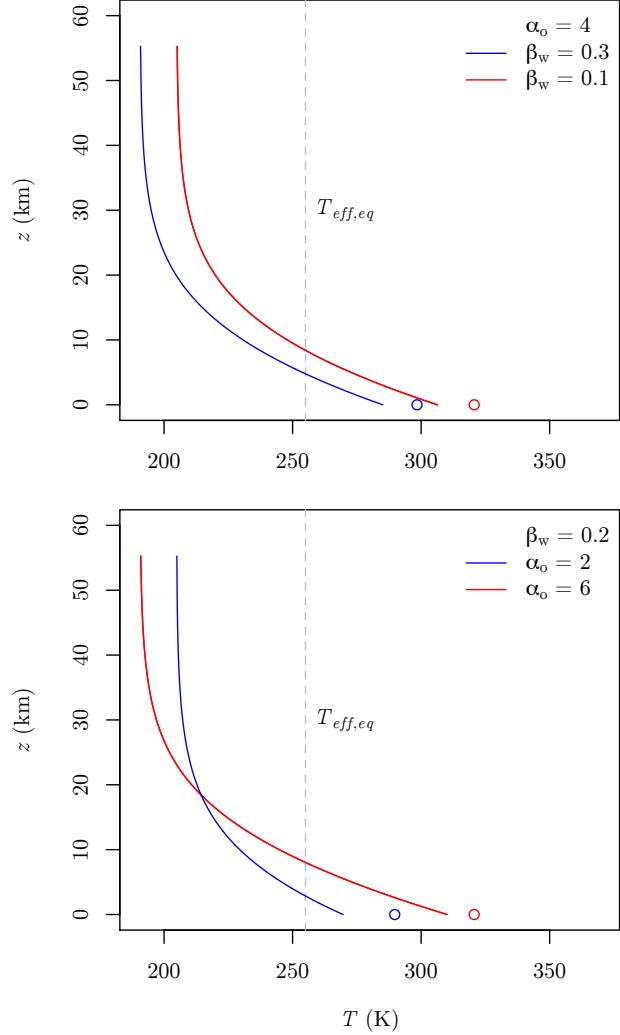

**Figure 7.** Vertical temperature profiles in overall equilibrium for the window-grey case with $T_{eff,eq} = 255\,\mathrm{K}$ for different combinations of the parameters $\alpha_o$ and $\beta_w$ (Eq. (30)). The circles at $z = 0$ denote the corresponding surface temperatures. Note that the vertical coordinate $z$ is only approximate height, calculated from $h$ with a constant scale height $H = 8\,\mathrm{km}$ such that $h = 1 - e^{-z/H}$.

and the temperature at the inner wall surface is somewhat lower than the interior (room) temperature. These temperature differences maintain an export of heat at the same rate at which the interior is heated. In the following we assume that the walls have negligible heat capacity whereas the interior reacts more inertly to disturbances due to a non-zero heat capacity.

We first assume that the building is insulated equally well everywhere, resulting in a uniform temperature of the outer surface.
If now the heat resistance of the walls is instantaneously increased, at first the outer surface temperature drops and the inner surface temperature rises, while the interior temperature is still unchanged. In this situation less heat escapes from the building



than is released by the heating system. The imbalance leads to a slow ascent of the interior temperature that continues until the outer surface temperature returns to its original value. The initial cooling of the outer surface temperature is analogous to the quasi-instantaneous (transient) cooling that occurs in the upper half of the atmosphere in the grey model.

Assuming instead that there are parts of the building envelope that are more weakly insulated than the remainder, as is typically the case with windows, the outer surface temperature in equilibrium is higher at the windows than it is at the walls, and a larger fraction of the total energy escapes via the windows compared to how much they contribute to the total area of the building envelope. If now the heat resistance of the walls is increased, the outer surface temperature of the wall is diminished not only temporarily, but some cooling remains also after the interior temperature has increased to its new equilibrium value. In the new equilibrium, even more energy escapes through the windows and less through the walls. The permanent cooling of the outer surface temperature of the walls is analogous to the cooling in the higher atmosphere associated with the permanent blocking effect of $CO_2$-induced MA cooling.

The main difference between the building analogy and the window-grey radiation model is that the separation between walls and windows in the former case is in geometrical space, whereas the separation into an opaque and a transparent radiation band in the latter case is in spectral space. Another obvious difference is that the mechanism of energy transfer is heat conduction in the walls of a building as opposed to radiation in the atmosphere. Nevertheless, we reckon the analogy of an insulated building as a valid means to illustrate the blocking effect of $CO_2$-induced MA cooling.

### 4.3 The indirect solar effect

On Earth not all solar radiation transects the air unhindered, but some is absorbed within the atmosphere and leads to increased temperatures, particularly in the upper parts of the ozone layer. The solar heating can be incorporated into Eq. (24) as an additional term $S^*(h)$:

$$2\epsilon_o \sigma T^*(h)^4 (1 - \beta_w) = \epsilon_o \left( O^\uparrow(h) + O^\downarrow(h) \right) + S^*(h) . \tag{36}$$

Eq. (36) is similar to Eq. (6.15) in Neelin (2011), except that Neelin considers only the grey case ($\beta_w = 0$) and neglects the downwelling LW radiation, constraining the validity of the equation to the vicinity of the TOA.

Assuming that solar heating is confined to an infinitesimally thin layer at $h = h'$, such that the equilibrium temperature everywhere else remains unchanged and, thus, $O^\uparrow(h')$ and $O^\downarrow(h')$ are not affected by the additional term, one arrives at

$$T^*(h')^4 = T(h')^4 + \frac{s^*(h')}{\alpha_o(1 - \beta_w)} , \tag{37}$$

where $T(h')$ is the solution of Eq. (36) with $S^*(h') = 0$, that is, the window-grey solution of Eq. (24), and $s^*(h') = S^*(h')/(2\sigma \, \mathrm{d}h)$.

Equation (37) reveals the following: Given that due to an additional term in the local energy budget the atmospheric temperature at some height is deflected from the window-grey solution, increasing the amount of LW absorbers in the atmosphere results in a relaxation of the temperature towards the window-grey solution. This holds both for increasing $\alpha_o$ and for decreasing $\beta_w$. It must be kept in mind though that the window-grey solution itself depends on $\alpha_o$ and $\beta_w$ (Eq. (30)), making the



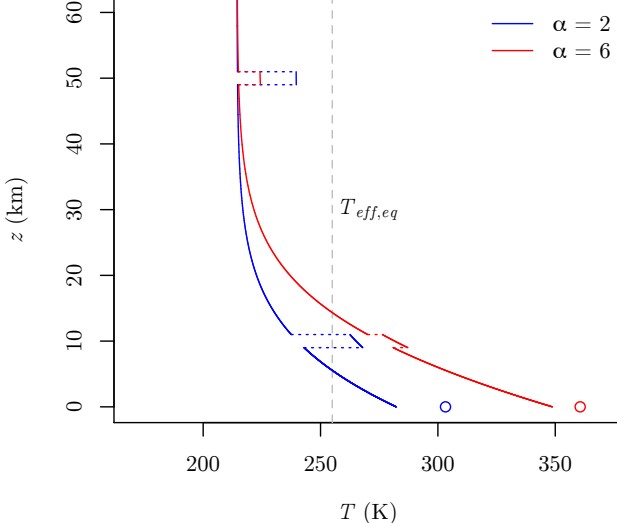

**Figure 8.** Vertical temperature profiles of a grey atmosphere that is additionally locally heated (e.g., by absorption of solar radiation) at two heights within the atmosphere. Apart from the heights at which the profiles are locally deflected due to additional heating, the blue and red curves show the same grey equilibria as the corresponding curves in Fig. 2. Again $\alpha$ corresponds to $\alpha_o$ in the window-grey model with $\beta_w = 0$. The heights at which additional heating occurs ($z_1 \approx 10\,\mathrm{km}$, $z_2 \approx 50\,\mathrm{km}$) and the magnitude of the additional heating terms (specified such that the temperature rise is $25\,\mathrm{K}$ for $\alpha = 2$ at both heights) are more or less arbitrarily chosen to demonstrate the effect. $T_{eff,eq} = 255\,\mathrm{K}$. The circles at $z = 0$ denote the corresponding surface temperatures. Note that the vertical coordinate $z$ is only approximate height, calculated from $h$ with a constant scale height $H = 8\,\mathrm{km}$ such that $h = 1 - e^{-z/H}$.

relaxation towards the window-grey solution an additional effect. Figure 8 illustrates the indirect solar effect for the grey case (i.e., for $\beta_w = 0$).

If the additional term $s^*$ is positive, as it is the case for the absorption of solar radiation by ozone, increasing the emissivity either by increasing $\alpha_o$ or by decreasing $\beta_w$ results in local cooling. We call this effect the *indirect solar effect* of $CO_2$-induced MA cooling. The indirect solar effect, like the permanent blocking effect, is still at work when the system has reached the new (more opaque) overall equilibrium.Note that the indirect solar effect would also manifest if the opacity was changed only locally. This is not the case for the blocking effect, where integration over a finite layer with perturbed opacity is needed.

## 5 Effect strengths

An essential question so far unanswered is how strong the above derived effects are compared to each other. In this section we first discuss what can, and what can not, be learnt from the window-grey model regarding the relative effect strengths. We then discuss results obtained with a complex atmospheric model that provide further quantitative evidence.



## 5.1 Preliminary considerations based on the window-grey model

Given its simplicity, quantitative statements based on the window-grey model are difficult to make. This is also why we have deliberately refrained from "tuning" the two model parameters to bring the model state close to the observed state of the Earth's atmosphere; the cases shown in Figs. 2, 5, 7, and 8 are quantitatively unrealistic.

5       To arrive at a reasonable vertical temperature profile, some simplifications would have to be relaxed, most importantly the assumption of vertically well-mixed greenhouse gases (violated in particular by water vapour), the simplistic LW band structure, and the neglect of vertical heat transport by convection (and conduction at the surface). Convection acts to set an upper limit to the (global-mean) lapse rate at ∼6.5 K, leading to an almost constant lapse rate in the troposphere (e.g., Manabe and Wetherald, 1967). The appearing radiative-convective equilibrium in the troposphere is associated with an upward heat

transport and increased temperatures in the free troposphere and decreased temperatures at (and close to) the surface.

The region where convection plays an important role (the troposphere) coincides with the region where water vapour is abundant and dominates the LW absorption spectrum of the atmosphere. The latter implies that the effective width of the atmospheric window is very different for the atmosphere as a whole ($\beta_w \approx 10 - 20\%$) and for the atmosphere beyond the tropopause alone ($\beta_w \approx 90 - 95\%$; the actual values depend on the optical thickness threshold used to derive $\beta_w$ from the

continuous absorption spectra, compare Fig. 1b-c). The width of the atmospheric window is however crucial for the atmospheric temperature profile and the strength of the MA cooling effects both in absolute and relative terms. One can derive from Eqs. (29) and (31) for the strength of the permanent blocking effect at the TOA in relation to the surface response that

$$\frac{\partial T_{toa,eq}}{\partial \alpha_o} \Big/ \frac{\partial T_{srf,eq}}{\partial \alpha_o} = -\frac{1}{2} \left( \frac{T_{srf,eq}}{T_{toa,eq}} \right)^3 \frac{\beta_w}{1-\beta_w} . \tag{38}$$

Inserting typical temperatures prevailing at the Earth's surface (∼290 K) and at the mesopause (∼180 K), the estimates of $\beta_w$

given above correspond to response ratios of approximately only $(-0.2) - (-0.5)$ if $\beta_w$ is representative of the whole atmosphere, but $(-20) - (-40)$ if $\beta_w$ is representative only of the largely water-free atmosphere beyond the tropopause (compare Fig. 1). To derive absolute estimates corresponding to the current atmospheric $CO_2$ perturbation, multiply by ∼1.5 K which is an estimate for the surface warming that would eventually occur if $CO_2$ concentrations were kept at current levels (this factor is consistent with an increase of $CO_2$ concentrations by a factor $\sqrt{2}$ since pre-industrial times from ∼280 ppmv to ∼400 ppmv,

combined with a climate sensitivity of 3 K; compare IPCC, 2013). An additional confounder to this wide range of estimates is the crude assumption that $CO_2$ only affects $\alpha_o$; in the less opaque spectral regions of its absorption bands, $CO_2$ affects $\beta_w$ rather than $\alpha_o$, thereby reducing the cooling (see also Appendix B).

A rough estimate for the cooling from the indirect solar effect can be obtained from Eq. (37) by (again) using the mesopause temperature (∼180 K) as substitute for the window-grey background solution $T(h')$. At the stratopause, where the solar heating

and thus the expected indirect solar effect is strongest, it is $T^*(h') \approx 270$ K. Again assuming that $\alpha_o$ has increased by a factor $\sqrt{2}$ since pre-industrial times, and given that $CO_2$ is the dominant greenhouse gas beyond the tropopause, the cooling at the stratopause caused by the indirect solar effect can be estimated from Eq. (37) to be ∼ −17 K (with the magnitude decreasing in both directions away from the stratopause). In this case it is not a confounder that an increase of $CO_2$ partly reduces the



effective width of the atmospheric window because it is the combined factor $\alpha_o(1 - \beta_w)$ that enters the indirect solar effect. Instead, uncertainties arise mainly from the crude estimate of the window-grey background solution and from the contribution of non-$CO_2$ greenhouse gases (including ozone) to $\alpha_o(1 - \beta_w)$, making the above estimate rather an upper bound of the actual effect strength.

Finally, the cooling from a transient blocking effect seen in the window-grey model before the surface has attained a new equilibrium is bounded by the remaining surface warming that would eventually occur if greenhouse gases were stabilised at current levels (see also Appendix B2). Assuming that $\sim 1.0\,\mathrm{K}$ of the $\sim 1.5\,\mathrm{K}$ equilibrium surface warming has already taken place (consistent with the above assumptions), at most $\sim 0.5\,\mathrm{K}$ of the present MA cooling could in principle be explained with a transient effect. The main caveat with this estimate appears to be that the transient effect derived above does not account for

additional tropospheric opacity changes that accompany the surface adjustment (linked to the water vapour feedback; see next section).

In summary, estimates based on the window-grey model indicate that the indirect solar effect might have caused up to $17\,\mathrm{K}$ of $CO_2$-induced cooling (at the stratopause) since pre-industrial times, whereas today's contribution of a transient blocking effect appears to be limited to half a degree cooling at the most (at all heights). By contrast, the window-grey model is not well

suited to constrain the strength of the permanent blocking effect because details of the LW absorption spectrum of both $CO_2$ and the other atmospheric greenhouse gases play a crucial role. According to these estimates, the indirect solar effect could in principle be strong enough to explain the recent observed cooling around the stratopause ($\sim -2\,\mathrm{K}$ per decade Beig, 2011; Huang et al., 2014). However, the fact that the observed cooling does not diminish strongly beyond the stratopause suggests an increasing contribution of the permanent blocking effect in the mesosphere, as confirmed in the following section.

## 5.2    Simulations with a complex atmospheric model

To complement the findings obtained with the window-grey model, and to derive more conclusive estimates for the strength of the effects, we have conducted simulations with the complex atmospheric general circulation model ECHAM6 (Stevens et al., 2013). This model and its predecessors have been used for comprehensive simulations, including future climate projections, in the different phases of the Coupled Model Intercomparison Project (CMIP), which are the backbone of the reports compiled

by the Intergovernmental Panel on Climate Chance (IPCC). These models, including ECHAM6, therefore have sophisticated parameterizations for, e.g., radiation, convection, clouds, boundary-layer turbulence, and gravity waves, and numerically solve the governing equations of fluid dynamics on grids with steadily increasing spatio-temporal resolution. The radiative transfer scheme used in ECHAM6, which employs 16 LW bands, has been shown to give instantaneous clear-sky responses to greenhouse-gas perturbations in close agreement with accurate line-by-line calculations (Iacono et al., 2008).

To adequately resolve the middle atmosphere, we have used the T63L95 configuration with relatively coarse ($\sim 2°$) horizontal but high (95 levels, top at 0.01 hPa) vertical resolution. The ocean and sea ice have been treated in a simple way similar to the approach of Dickinson et al. (1978). Each year (Jan1–Dec31) the ocean surface temperature and sea ice concentration and thickness are prescribed with a realistic seasonal and spatial pattern derived from observations. After every year the ocean surface temperature pattern is updated uniformly according to the total energy imbalance integrated over the global



**Table 1.** ECHAM6 simulations

| Simulation ID | Solar absorption in the atmosphere | SST treatment | $CO_2$ [ppmv] | CFC-11 / -12 [ppbv] | GMST[a] |
|---|---|---|---|---|---|
| REF | yes | free equilibration | 280 | 0.2528 / 0.4662 | 286.8 |
| $CO_2 x2$ | yes | free equilibration | 560 | 0.2528 / 0.4662 | 286.8+2.62 |
| $CO_2 x2_{\text{fixSST}}$ | yes | prescribed from REF | 560 | 0.2528 / 0.4662 | 286.8+0.28 |
| $CFC x15$ | yes | free equilibration | 280 | 3.792 / 6.993 | 286.8+2.20 |
| $CFC x15_{\text{fixSST}}$ | yes | prescribed from REF | 280 | 3.792 / 6.993 | 286.8+0.15 |
| $REF_{\text{ns}}$ | no | free equilibration | 280 | 0.2528 / 0.4662 | 285.0 |
| $CO_2 x2_{\text{ns}}$ | no | free equilibration | 560 | 0.2528 / 0.4662 | 285.0+2.20 |
| $CFC x15_{\text{ns}}$ | no | free equilibration | 280 | 3.792 / 6.993 | 285.0+1.74 |

[a] global annual-mean near-surface temperature

ocean surface (including sea ice) and over the year, using a heat capacity that corresponds to a 50 m thick mixed-layer ocean. Despite changing temperatures, the sea ice state pattern is not updated, leading to discrepancies between the sea ice and ocean states. This procedure also suppresses further changes to the surface temperature pattern, such as polar warming amplification. However, this allows for a rapid thermal equilibration of the surface with an exponential timescale of ∼3 years, serving the

purpose of this paper where the focus is on the global-mean response.

    We have conducted eight ECHAM6 simulations with differences in (i) the treatment of solar radiation, (ii) the treatment of sea-surface temperatures (SST), and (iii) the abundance of greenhouse gases (Tab. 1). In five simulations the solar radiation follows the default behavior, with some of the solar radiation absorbed within the atmosphere. This set includes one reference simulation where pre-industrial greenhouse-gas concentrations are used and the SSTs are allowed to run into equilibrium, and

four sensitivity simulations. In two of these the ocean is allowed to attain a new equilibrium, either with the $CO_2$ concentration doubled or with the Chlorofluorocarbon (CFC-11 and CFC-12) concentrations increased by the factor 15, chosen such that the surface warming is similar compared to the case of $CO_2$ doubling. The other two sensitivity simulations are identical with the previous two, except that the SSTs are prescribed from the reference simulation.

    In another set of three simulations the absorption of solar radiation within the atmosphere is turned off. This set also includes

a reference simulation and two sensitivity simulations with increased $CO_2$ and CFC concentrations. In these, the SSTs are again allowed to run into equilibrium. All simulations are conducted over 18 years, but only the last 10 years are used to compute averages for the analysis because it takes a few years (in our setup) until an equilibrium is reached.

    This experimental design allows us first to investigate the dependence of MA temperature changes on the spectral properties of the added absorbers: CFCs absorb mainly in the spectral window of the Earth's atmosphere, whereas $CO_2$ absorbs mainly

at wavelengths where the atmosphere is already relatively opaque. Second, we can quantify the effect strength for the two permanent effects by which $CO_2$ cools the MA, deduced above with the window-grey model, and investigate how atmospheric temperatures respond to the slow surface adjustment.





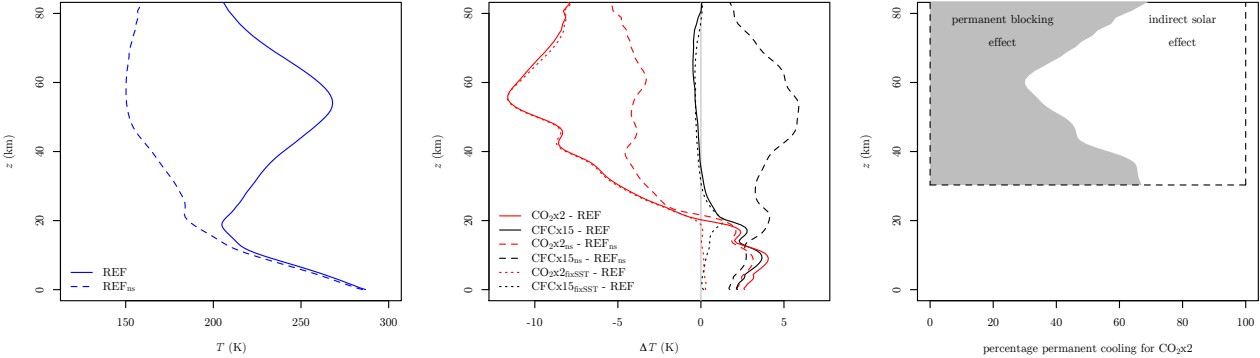

**Figure 9.** Results obtained with ECHAM6 coupled to a simplistic ocean model to allow for rapid thermal adjustment of the surface. Left: Global annual-mean equilibrium temperature profiles for two reference runs under pre-industrial external forcing, with (solid; REF) and without (dashed; $REF_{ns}$) absorption of solar radiation in the atmosphere. Middle: Temperature difference to the corresponding reference runs in response to increased $CO_2$ or CFC concentrations (simulation IDs are explained in Tab. 1). Right: Percentage of the permanent cooling effect in response to $CO_2$ doubling from the window and indirect solar effects, estimated by dividing $CO_2x2_{ns} - REF_{ns}$ by $CO_2x2 - REF$. Note that the vertical coordinate $z$ is only approximate height, calculated from $h$ with a constant scale height $H = 8$ km such that $h = 1 - e^{-z/H}$.

With standard treatment of solar radiation, $CO_2$ doubling in ECHAM6 leads to an increase in global annual-mean near-surface temperature by 2.6 K (Fig. 9 middle, red solid curve); this is the climate sensitivity of our model setup. The tropospheric warming in fact increases with height, reaching a maximum of 4.1 K in the upper troposphere. This pattern, which is not captured by the window-grey model, results from the temperature dependence of the moist-adiabatic lapse rate (the so-called lapse-rate feedback) and is thus related to convective processes. Slightly above the tropopause the temperature response changes sign and assumes a maximum cooling by 11.6 K around the stratopause region.

Adding CFCs instead of $CO_2$ results (by design) in a similar tropospheric response with a near-surface warming by 2.2 K, but temperatures in the MA remain virtually unchanged (Fig. 9 middle, black solid curve). The indirect solar effect is not wavelength dependent: more absorbers increase emission more strongly than they increase absorption, thereby reducing the relative importance of the solar heating term (Sects. 4.3 and 5.1). This suggests that another effect counteracts the cooling from the indirect solar effect. The above considerations based on the window-grey model suggest that this counteracting warming effect can be interpreted as an *inverse blocking effect*: Instead of making the already opaque part of the spectrum even more opaque, which mainly happens when $CO_2$ is added, the increase of CFC concentrations acts to narrow the atmospheric window, corresponding to a decrease of $\beta_w$ in the window-grey model (Figs. 7 and 6 top). In fact, the situation corresponds not only to a decrease of $\beta_w$, but also to a simultaneous decrease of $\alpha_o$ because the average opacity of what should be translated into the





single opaque band of the window-grey model is decreased by the inclusion of the CFC-affected – still relatively transparent – parts of the previous window band. Overall, the MA is more strongly subjected to the radiation from the warm surface.

When solar radiation passes the model atmosphere unhindered, tropospheric temperatures are slightly reduced by $\sim 2\,\mathrm{K}$ (Tab. 1 and Fig. 9 left). More importantly, the local temperature maximum around the stratopause completely disappears and
temperatures drop to $\sim 160\,\mathrm{K}$ in the upper stratosphere and in the mesosphere. Adding $CO_2$ or CFCs under these modified background conditions results in similar responses in the troposphere, but markedly different responses in the MA (Fig. 9 middle, dashed curves): While the cooling in response to $CO_2$ is roughly halved, the previously neutral response to CFCs turns into a substantial warming by up to 5.9 K. These results are consistent with the interpretation that the cooling due to the indirect solar effect has been precluded, leaving only the response due to the blocking effect (in the $CO_2$ case) and the inverse blocking
effect (in the CFC case).

Under the assumption of linearity, this allows us to estimate the fractional contributions of the two permanent effects to MA cooling (Fig. 9 right). According to our results the indirect solar effect contributes up to $\sim 70\%$ to the total permanent cooling around the stratopause where solar heating is strongest. Outside this region the blocking effect gains importance and begins to dominate the cooling in the middle stratosphere and the middle mesosphere. The assumption of linearity is rather crude, so
these estimates should be taken with a grain of salt. In fact, it is probably not possible to make a completely clean quantitative distinction, as the formal analysis in Appendix B3 suggests.

Finally, the window-grey model also suggests a transient MA cooling that adds to the permanent cooling before the surface temperature has adjusted to the changed radiative forcing. We can investigate this effect with the remaining two simulations where the greenhouse gases are perturbed but SSTs are fixed to the reference state (Tab. 1). Interestingly, the initial MA
responses (Fig. 9 middle, dotted curves) are nearly identical with the corresponding equilibrium responses (solid curves) above $\sim 20\,\mathrm{km}$. This means that, given a fixed atmospheric composition in terms of well-mixed greenhouse gases, MA temperatures are quasi independent of the surface temperature. This is not obvious as a warming of the surface and troposphere should lead to increased upwelling LW radiation.

A possible explanation for the absence of a visible transient effect is that the water vapor feedback – higher tropospheric
temperatures imply higher water vapor concentrations – leads to a pronounced temperature dependence of the atmospheric opacity. This effect is neglected in the window-grey model presented above. The increased opacity as a result of tropospheric warming entails that a *secondary blocking effect* counteracts the slow reduction of the initial MA cooling associated with the transient blocking effect seen in the window-grey model.

Because of the stronger $CO_2$ perturbation (factor 2 versus factor $\sim\sqrt{2}$), the estimates in this section must be divided ap-
proximately by 2 for comparability with the estimates in Sect. 5.1 where temperature changes between pre-industrial times and today are discussed. Obviously, the strength of the indirect solar effect inferred from the window-grey model is considerably overestimated, even when compared to the total simulated cooling at the stratopause (including a contribution from the permanent blocking effect).



# 6 Summary and conclusions

In this article we explain a well-known phenomenon that is central to our general understanding of climate change – cooling of the middle atmosphere (MA) by $CO_2$ – in the probably simplest possible way that is, in its essence, still physically correct. We do so by applying a vertically continuous window-grey radiation model to the phenomenon. This way it is possible to distinguish two main effects by which $CO_2$ cools the MA.

First, enhanced blocking of upwelling LW radiation operates towards lower MA temperatures. In principle, this *blocking effect* has a transient component due to the slow warming of the surface. This adjustment leads to intensified upwelling LW radiation and tends to reduce the initial MA cooling in the window-grey model. While these effects exactly compensate each other in a grey atmosphere, leading to an equilibrium skin temperature that is independent of the atmospheric opacity, the blocking of upwelling LW radiation outweighs in the presence of a spectral window because of the reduced surface temperature sensitivity, leaving lower equilibrium temperatures above a critical height after the adjustment. Hence, the blocking effect is permanent because the Earth's atmosphere is not grey, i.e., uniformly opaque for LW radiation at any wavelength, but absorbs and emits LW radiation with varying intensity depending on wavelength. The introduction of a spectral window into an otherwise uniformly opaque atmosphere is the simplest possible means to capture the effect in a physical model.

The second permanent effect of $CO_2$-induced MA cooling is the *indirect solar effect*. It owes its existence to the fact that there are heat sources within the atmosphere in addition to LW radiation, most importantly solar radiation that is absorbed in particular in the vicinity of the stratopause by ozone. The additional heating term causes a deviation of the temperature profile from the window-grey solution. The strength of this deviation depends on the abundance of LW absorbers because the relative importance of the constant additional heating term in the local energy budget decreases with increasing LW absorber abundance.

While the window-grey model allows for a fully analytical treatment of $CO_2$-induced MA cooling, it is not well suited to constrain the relative effect strengths. Uncertainties are large because the window-grey model entails a number of gross simplifications, including in particular: the assumption of vertically well-mixed greenhouse gases (violated in particular by water vapour); the simplistic LW band structure; and the neglect of vertical heat transport by convection (and conduction at the surface). Additional simplifications are: the neglect of Wien's law; the two-stream approximation; the neglect of the horizontal dimensions and the associated differential heating and atmospheric dynamics (including gravity waves); the neglect of chemical processes; the implicit treatment of solar radiation; the neglect of clouds, aerosols, and scattering in general; and the assumption of local thermodynamic equilibrium that does not hold in the upper mesosphere and beyond. Most of these factors are discussed for example in Pierrehumbert (2010), and those specific to the mesosphere are reviewed in Mlynczak (2000).

Therefore, to quantify the effect strengths and to complement the insights gained from the window-grey model, we have conducted simulations with a much more complex atmospheric model. The results indicate that the two permanent effects are similarly important, with the indirect solar effect dominating around the stratopause and the blocking effect dominating away from the stratopause. The window-grey model also predicts a slow (re-)warming throughout the atmosphere in response to the


slow surface warming. However, this transient effect is negligible in the MA according to the simulations with the complex model. A possible explanation is that additional MA cooling by a secondary blocking effect, caused by increasing tropospheric water vapour concentrations in the course of the surface adjustment, compensates for the slow decay of the transient MA cooling.

This article is meant to consolidate our understanding of why $CO_2$ cools the middle atmosphere by filling a gap between reality and complex atmospheric models on the one side and somewhat scattered heuristic arguments on the other. The reconsideration of $CO_2$-induced MA cooling based on the window-grey radiation model as put forward here has a distinct educational element, with the potential to convey the physical essence of this important phenomenon to a broader audience.

**Appendix A:  Relation to discrete-layer models**

Without showing derivations we point out that the vertically continuous model(s) presented in the main text can be interpreted as a generalization of discrete-layer models. The simplest type of the latter, a model with only one grey atmospheric layer, is widely used to explain the greenhouse effect in a conceptual way (e.g., Pierrehumbert, 2010; Neelin, 2011). In the following we discuss only the grey case, but the window-grey case can be treated analogously.

In an $N$-layer grey-atmosphere model with uniform layer emissivity $\epsilon_l$, from the radiative balances at every atmospheric
layer it follows that, given an arbitrary surface temperature, the equilibrium temperature at layer $i$ is

$$T_i = T_{srf} \sqrt[4]{\frac{\epsilon_l (n - i) + 1}{\epsilon_l (n - 1) + 2}} \tag{A1}$$

where $i = 1$ is the lowest and $i = n$ the highest atmospheric layer. The overall equilibrium situation is obtained when $T_{srf}$ in Eq. (A1) is replaced by the value it attains in overall equilibrium, which is

$$T_{srf,eq} = T_{eff,eq} \sqrt[4]{1 + \frac{n \epsilon_l}{2 - \epsilon_l}} . \tag{A2}$$

For $\alpha/2 \in \mathbb{N}$ the vertically continuous grey model is equivalent to a discrete grey model with $n = \alpha/2$ atmospheric layers, each with emissivity $\epsilon_l = 1$. The heights $h$ that correspond to the discrete levels $i$ are then determined by

$$h_i = \frac{i - \frac{1}{2}}{n} . \tag{A3}$$

Although providing a very suitable conceptual tool to understand the greenhouse effect, the discrete-layer grey-atmosphere model (just like its continuous analogue) obviously can not explain greenhouse-gas induced MA cooling. For such an expla-
nation it is again necessary either to introduce non-uniform opacity for LW radiation (e.g., by introducing an atmospheric window), to introduce an additional (solar) heating term, or to consider the transient phase.

**Appendix B:  Formal response analysis**

To supplement the discussion in the main text, in this appendix we quantify the response of temperature to changes in the parameters of the vertically continuous window-grey atmosphere model in terms of partial derivatives. We thereby also separate





the simultaneously occurring effects of $CO_2$-induced MA cooling in a formal way. We start without the indirect solar effect but include it into the formalism later.

In the following a *response* is simply the partial derivative of temperature with respect to either $\alpha_o$ or $\beta_w$. Different responses are discerned based on the conditions introduced into the derivatives. We distinguish between a fast (quasi-instantaneous) response $\mathcal{F}$ where the surface temperature is kept fixed at its previous equilibrium value, and a subsequent slow response $\mathcal{S}$ during which also the surface attains its new equilibrium temperature. The overall equilibrium response $\mathcal{E}$ can thus be written as

$$\mathcal{E} = \mathcal{F} + \mathcal{S}. \tag{B1}$$

During the slow transition from $\mathcal{F}$ to $\mathcal{E}$, the current response $\mathcal{C}(t)$ at time $t$ deviates from $\mathcal{E}$ by the transient response $\mathcal{T}(t)$:

$$\mathcal{C}(t) = \mathcal{E} + \mathcal{T}(t), \tag{B2}$$

with

$$\mathcal{T}(t) = \big(f(t) - 1\big)\mathcal{S}, \tag{B3}$$

where $f(t) \in [0,1]$ is that fraction of the slow response that has already taken effect at time $t$, with $f(0) = 0$ and $f(t \to \infty) = 1$. The transient response is thus defined as the part of the quasi-instantaneous response that is later compensated by the adjustment to surface warming.

## B1 Surface response

Differentiating Eq. (29) with respect to $\alpha_o$ and $\beta_w$ gives the overall equilibrium responses $\mathcal{E}_{\alpha_o}$ and $\mathcal{E}_{\beta_w}$ at the surface:

$$\mathcal{E}_{\alpha_o,srf} \equiv \frac{\partial T_{srf,eq}}{\partial \alpha_o} = \frac{T_{eff}}{4} \sqrt[4]{\frac{\alpha_o + 2}{\alpha_o \beta_w + 2}} \left( \frac{1}{\alpha_o + 2} - \frac{\beta_w}{\alpha_o \beta_w + 2} \right) \tag{B4}$$

and

$$\mathcal{E}_{\beta_w,srf} \equiv \frac{\partial T_{srf,eq}}{\partial \beta_w} = \frac{-T_{eff}\,\alpha_o}{4} \sqrt[4]{\frac{\alpha_o + 2}{(\alpha_o \beta_w + 2)^5}}. \tag{B5}$$

Excluding the trivial cases $\alpha_o = 0$ and $\beta_w = 1$, Eqs. (B4) and (B5) imply that $\mathcal{E}_{\alpha_o,srf} > 0$ and $\mathcal{E}_{\beta_w,srf} < 0$. That is, the surface warms when greenhouse gases are added.

As there is by definition no fast response at the surface, i.e., $\mathcal{F}_{\alpha_o,srf}, \mathcal{F}_{\beta_w,srf} = 0$, it is $\mathcal{S}_{\alpha_o,srf} = (f(t) - 1)\mathcal{E}_{\alpha_o,srf}$ and $\mathcal{S}_{\beta_w,srf} = (f(t) - 1)\mathcal{E}_{\beta_w}$ the transient response fully compensates for the equilibrium response initially (where $f = 0$), but vanishes for $t \to \infty$.



## B2  Atmospheric response

The temperature response of the continuous window-grey atmosphere in overall equilibrium to $\alpha_o$ and $\beta_w$ as a function of height is obtained by differentiating Eq. (30) with respect to the two model parameters, giving

$$\mathcal{E}_{\alpha_o}(h) \equiv \frac{\partial T_{eq}(h)}{\partial \alpha_o} = \frac{T_{eff,eq}}{4} \sqrt[4]{\frac{\alpha_o(1-h)+1}{\alpha_o\beta_w+2}}$$
$$\cdot \left( \frac{1-h}{\alpha_o(1-h)+1} - \frac{\beta_w}{\alpha_o\beta_w+2} \right) \tag{B6}$$

and

$$\mathcal{E}_{\beta_w}(h) \equiv \frac{\partial T_{eq}(h)}{\partial \beta_w} = \frac{-T_{eff,eq}\,\alpha_o}{4} \sqrt[4]{\frac{\alpha_o(1-h)+1}{(\alpha_o\beta_w+2)^5}} \,. \tag{B7}$$

Differentiating the quasi-instantaneous temperature profile given by Eq. (26) in overall equilibrium at $h = 1$ (i.e., at the TOA) with respect to $\alpha_o$ leads to a form that supports the interpretation of the permanent blocking effect as the interplay between the sensitivity of the surface temperature to greenhouse-gases on the one hand and the blocking of upwelling LW radiation by greenhouse gases on the other hand:

$$\mathcal{E}_{\alpha_o,toa} = \sqrt[4]{\frac{1}{\alpha_o+2}} \left( \frac{\partial T_{srf,eq}}{\partial \alpha_o} - \frac{T_{srf,eq}}{4(\alpha_o+2)} \right) \,. \tag{B8}$$

Here the surface sensitivity is represented by the minuend in the brackets whereas the blocking effect is represented by the subtrahend in the brackets.

Differentiating Eq. (26) with respect to $\alpha_o$ under the constraint $T_{srf} = $ const and inserting Eq. (29) gives the fast temperature response as a function of height:

$$\mathcal{F}_{\alpha_o}(h) \equiv \frac{\partial T(h)}{\partial \alpha_o} \bigg|_{T_{srf}=T_{srf,eq}=\text{const}}$$
$$= \frac{T_{eff,eq}}{4} \sqrt[4]{\frac{\alpha_o(1-h)+1}{\alpha_o\beta_w+2}} \left( \frac{1}{\alpha_o+(1-h)^{-1}} - \frac{1}{\alpha_o+2} \right) \,. \tag{B9}$$

Note that changing the width of the atmospheric window entails no fast response, i.e., $\mathcal{F}_{\beta_w}(h) = 0$.

The transient part of the response follows from Eqs. (B6) and (B9) with Eqs. (B1)–(B3) as

$$\mathcal{T}_{\alpha_o}(h,t) \equiv \big(1-f(t)\big)\big(\mathcal{F}_{\alpha_o}(h) - \mathcal{E}_{\alpha_o}(h)\big)$$
$$= \big(1-f(t)\big)\frac{T_{eff,eq}}{4} \sqrt[4]{\frac{\alpha_o(1-h)+1}{\alpha_o\beta_w+2}}$$
$$\cdot \left( \frac{\beta_w}{\alpha_o\beta_w+2} - \frac{1}{\alpha_o+2} \right) \,. \tag{B10}$$

The transient part of the response therefore becomes smaller with height. Comparison with Eq. (B4) further shows that $|\mathcal{T}_{\alpha_o}(h,t)| < |\mathcal{E}_{\alpha_o,srf}|$, that is, the transient cooling at any height in the atmosphere is always weaker than the equilibrium warming of the surface. Note that $\mathcal{T}_{\beta_w}(h,t)$ follows directly from $\mathcal{E}_{\beta_w}(h)$ because $\mathcal{F}_{\beta_w}(h) = 0$.





**B3   Inclusion of the indirect solar effect**

Considering overall equilibrium, and simplifying the annotation by leaving away $h'$, differentiation of Eq. (37) with respect to the two model parameters gives

$$\mathcal{E}^*_{\alpha_o} \equiv \frac{\partial T^*_{eq}}{\partial \alpha_o} = \left( \frac{T_{eq}}{T^*_{eq}} \right)^3 \mathcal{E}_{\alpha_o} - \frac{s^*}{4 T^{*\,3}_{eq} \alpha_o^2 (1 - \beta_w)} \tag{B11}$$

and

$$\mathcal{E}^*_{\beta_w} \equiv \frac{\partial T^*_{eq}}{\partial \beta_o} = \left( \frac{T_{eq}}{T^*_{eq}} \right)^3 \mathcal{E}_{\beta_o} + \frac{s^*}{4 T^{*\,3}_{eq} \alpha_o (1 - \beta_w)^2} \, , \tag{B12}$$

where $\mathcal{E}_{\alpha_o}$ and $\mathcal{E}_{\beta_w}$ are the window-grey overall equilibrium responses given by Eqs. (B6) and (B7).

To include the indirect solar effect $\mathcal{I}$ into the formalism of Eqs. (B1)–(B3), one can extend Eq. (B2) using Eqs. (37), (B11), and (B12) as follows:

$$C^*(t) = \underbrace{\mathcal{E} + \mathcal{X}_{\mathcal{E}\mathcal{I}} + \mathcal{I}}_{\mathcal{E}^*} + \underbrace{\mathcal{T}(t) + \mathcal{X}_{\mathcal{T}\mathcal{I}}(t)}_{\mathcal{T}^*(t)} \tag{B13}$$

with

$$\mathcal{I}_{\alpha_o} = - \frac{s^*}{4 T^{*\,3}_{eq} \alpha_o^2 (1 - \beta_w)} \, , \tag{B14}$$

$$\mathcal{I}_{\beta_w} = \frac{s^*}{4 T^{*\,3}_{eq} \alpha_o (1 - \beta_w)^2} \, , \tag{B15}$$

$$\mathcal{X}_{\mathcal{E}\mathcal{I}} = \left[ \left( \frac{T_{eq}}{T^*_{eq}} \right)^3 - 1 \right] \mathcal{E} \, , \tag{B16}$$

$$\mathcal{X}_{\mathcal{T}\mathcal{I}}(t) = \left[ \left( \frac{T_{eq}}{T^*_{eq}} \right)^3 - 1 \right] \mathcal{T}(t) \, , \tag{B17}$$

where the terms in Eqs. (B14)–(B16) follow naturally from Eqs. (B11) and (B12). Equation (B17) results from Eq. (B13) with the analogues of Eqs. (B11) and (B12) for the fast response (i.e., with $\mathcal{E}^*$ and $\mathcal{E}$ replaced by $\mathcal{F}^*$ and $\mathcal{F}$) and the definition of $\mathcal{T}^*(t)$:

$$\mathcal{T}^*(t) = (1 - f(t))(\mathcal{F}^* - \mathcal{E}^*) \, . \tag{B18}$$

The terms $\mathcal{X}_{\mathcal{E}\mathcal{I}}$ and $\mathcal{X}_{\mathcal{T}\mathcal{I}}$ are interaction (or synergy) terms that result from the fact that $\mathcal{E}$, $\mathcal{I}$, and $\mathcal{T}(t)$ are not linearly additive. Due to these terms, the quantitative attribution of a total response to the different mechanisms is not unambiguously possible.

**Appendix C:  Supporting information on Figure 1**

The absorption spectra have been computed with *HITRAN on the Web* (http://hitran.iao.ru). The atmosphere from surface to space (Fig. 1b) was approximated with a 8,000 m thick homogeneous gas mixture at 260 K and 1,013.25 hPa with the following



composition (with respect to volume): 4,000 ppm $H_2O$, 300 ppm $CO_2$, 0.4 ppm $O_3$, 0.3 ppm $N_2O$, 1.7 ppm $CH_4$, 209,000 ppm $O_2$, and the remainder $N_2$. The middle atmosphere (Fig. 1c) was approximated with a 8,000 m thick homogeneous gas mixture at 220 K and 202.65 hPa with the same composition except for $H_2O$ (4 ppm) and $O_3$ (2 ppm) (and correspondingly $N_2$). Some Gaussian smoothing was applied to the spectra.

5 *Acknowledgements.* The foundations for this work have been laid during the authors' employment at the Max Planck Institute for Meteorology, Hamburg, Germany. The ECHAM6 simulations have been conducted at the German Climate Computing Centre (DKRZ). We thank Stephan Bakan and Thomas Rackow for helpful discussions.



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
