# Peer review of "Why $CO_2$ cools the middle atmosphere – a consolidating model perspective"

_Earth System Dynamics, 2016_

## Short Comment (SC1) · 22 Mar 2016

A study of this sort is sorely needed. As the authors identify in their introduction, the literature is lacking a clear explanation of why CO2 increases cool the middle atmosphere. Most explanations are either too simplistic to be accurate, or arise from complex model calculations so their physical interpretation is unclear. The manuscript is very clear and well-organised. I also appreciated the 'building' analogy for understanding the results. I have a few comments and suggestions.

P2, L9-10: " As ozone concentrations are expected to recover in future, it seems likely that CO2 concentrations will be of growing importance also in the lower stratosphere." I find Cordero and Forster (2006) an unusual reference here as they don't seem to present model results for the 21st Century, during which ozone concentrations are expected to recover. Model results for this period have been presented by, e.g. Stolarski et al (2008) and Ferraro et al (2015).

P5, L8 and L10: You seem to switch notation for optical depth from delta to tau here.

P10, L3: I am a little confused by your reference to "skin temperature" here. Equation 31 is for TOA temperature. I interpret skin temperature to mean the temperature of the surface.

Figure 9: I find the text on the legend and axes labels a bit small here. Perhaps you might consider increasing the font size?

Ferraro, A., Collins, M. and Lambert, F. (2015). A hiatus in the stratosphere?. Nature Climate Change, 5(6), pp.497-498, doi:10.1038/nclimate2624.

Stolarski, R., Douglass, A., Newman, P., Pawson, S. and Schoeberl, M. (2010). Relative Contribution of Greenhouse Gases and Ozone-Depleting Substances to Temperature Trends in the Stratosphere: A Chemistry–Climate Model Study. Journal of Climate, 23(1), pp.28-42, doi:10.1175/2009JCLI2955.1.

---

## Referee Comment (RC1) · M. Popp (Referee) · 9 May 2016

**Assessment of the manuscript**

The study aims at giving an educational perspective of the stratospheric cooling under increased CO$_2$ concentrations. The paper is somewhat unique in that it is neither a classical research paper that puts forward new results or a new piece of theory, nor a review paper that purely synthetizes past findings. Instead the paper uses a simple model to illustrate different effects that contributes to the stratospheric cooling under increased CO$_2$ concentrations. The results obtained with this simple model are not exceedingly interesting for people who worked with such models before. However, as pointed out by the authors, a publication with such a simple model on the topic is lacking

and therefore the approach to the problem can be considered novel. The study will thus certainly be very useful to scientists that are interested in the topic but not particularly familiar with radiative transfer, especially since the manuscript is well written and easy to follow. The study, however, still lacks a detailed discussion of how a convectively dominated troposphere affects the stratospheric cooling under increased $CO_2$ concentrations. Without such a discussion, it is hard to evaluate in how far the effects described with the local-radiative-equilibrium model are responsible for the stratospheric cooling in comprehensive models. Therefore, I recommend publication after revising the manuscript to this end.

**General comments**

The study makes use of a radiative-equilibrium model for the entirety of the atmosphere. These model are usually rather referred to as "local-radiative-equilibrium models" than as "radiative-equilibrium models", to indicate that the atmosphere is in radiative equilibrium everywhere. Such models give good results for optically thin regions of the atmosphere. However, the quality of results is difficult to assess in regions that are strongly influenced by convection such as Earth's troposphere. Changes in Earth's troposphere will most certainly also affect the response of the stratosphere to an increase in $CO_2$. Therefore, the manuscript would be strengthened, if the local-radiative-equilibrium model was discussed in more detail in the context of present day Earth, and if the role of the troposphere in cooling the stratosphere was quantitatively addressed.

Here are a few suggestions on how this could be done and for some potentially interesting aspects to discuss:

An easy way of incorporating the troposphere into the gray version of the model is to define the tropopause at the level $h = h_{TP}$. Applying this definition to equation (8) of

the manuscript yields

$$L^\uparrow_{TOA} = \frac{L^\uparrow_{TP}}{\frac{\alpha}{2}(1 - h_{TP}) + 1} \quad , \tag{1}$$

where $L^\uparrow_{TP}$ is the upward longwave flux at the tropopause. Applying this expression for $L^\uparrow_{TOA}$ to equation (10) of the manuscript yields

$$T(h) = \sqrt[4]{\frac{L^\uparrow_{TP}}{\sigma} \frac{\alpha(1 - h) + 1}{\alpha(1 - h_{TP}) + 2}} \quad , \tag{2}$$

or in the case with a window

$$T(h) = \sqrt[4]{\frac{O^\uparrow_{TP}}{(1 - \beta_W)\sigma} \frac{\alpha_O(1 - h) + 1}{\alpha_O(1 - h_{TP}) + 2}} \quad . \tag{3}$$

These equations can now serve as a starting point for the investigation of the influence of the upwelling longwave-radiation from the troposphere on the stratospheric response to an increase in $CO_2$ concentrations. This formulation decouples the stratosphere from the surface temperature and allows for different assumptions on how the changes at the surface impact the upwelling longwave radiation at the tropopause. A potentially interesting case to study would for example be the one of constant tropopause temperature, which is a common feature in climate-change simulations. Another interesting consideration would be, how the deepening of the troposphere and hence the increase of $h_{TP}$ influences the results. The authors could also apply some of the values they obtained from the simulations with ECHAM6 to $L^\uparrow_{TP}$, $O^\uparrow_{TP}$ $h_{TP}$, $T_{TP}$ ect..

A discussion of the troposphere would furthermore allow to discuss changes in albedo $A$ through changes in clouds. With $A$ included, the TOA radiative balance simply writes as

$$L^\uparrow_{TOA} = (1 - A)S \quad . \tag{4}$$
Changes in A would also lead to a change in the steady-state stratospheric temperature even in the gray case without window.

**Specific comments**

**Page 2, Line 27:** It would be helpful for interested readers, if the relevant Chapter in Goody and Yung (1989) would be indicated.

**Page 4, equation (1):** It may be worth mentioning, that equation (1) is obtained by making an assumption on mean angle of the thermal radiation against the vertical.

**Page 5, Line 15:** Equation (3) is the spectrally integrated, gray-absorption-case of Schwarzschild's equation.

**Page 6, Lines 14-17:** The last sentence of this paragraph suggests that the local radiative equilibrium is a decent approximation for present-day Earth. This is generally not the case. Despite sharp gradients existing close to the surface on Earth, the discontinuity and the lapse rate of the local-radiative-equilibrium model are by far too large. I suggest mentioning this.

**Page 10:** I suggest to introduce equation (35) just after equation (31), because equation (35) was quite important for me to understand the following discussion.

**Page 10, Line 17 to Page 11, Line 1:** This is one of the situations where a comparison between the local-radiative-equilibrium model and Earth is awkward. The stratosphere and the tropopause do not adjust at the same time-scale.

**Page 11, Line 26:** It may be worth mentioning that this is entirely a consequence of the outgoing longwave radiation (OLR) having to balance S, which is constant in steady state. As a consequence, even in this simple model a solar and a greenhouse forcing act differently: An increase in S would force the OLR to increase as well and,

as a consequence, the Skin-temperature would have to increase. In contrast, with this simple model the OLR does not change when $CO_2$ is applied.

**Page 12, Line 11:** The definition of "blocking effect" may be a bit confusing, because the authors already discussed the "increased blocking" in the window-less case. Speaking of "transient blocking effect" and "equilibrium blocking effect" may be a possible way of resolving the possible confusion. Note, that it is also arguable that in this case the "blocking effect" applies to present-day Earth, because the surface temperature on Earth has not equilibrated yet.

**Page 17, Line 8:** 6,5 K / 1000 m instead of 6,5 K.

**Page 17, Lines 7-9:** I am not sure whether it can be stated that there is an upper limit of the lapse rate of 6,5 K / 1000m. If, for example, the global-mean surface-temperature would change, the lapse rate may change as well. The fact, that the global-mean lapse rate is 6,5 K / 1000 m, does not necessarily mean that this value is a an upper global-mean limit.

**Page 17, Lines 11-27:** I suggest replacing this paragraph, by a discussion along the lines made in the general comments. I doubt that much meaningful can be learnt from applying typical values for present-day Earth to the local-radiative-equilibrium model without decoupling the surface-temperature from the TOA-temperature to some degree.

**Page 18, Line 19:** I suggest replacing "confirmed" by "supported". Model results are no observations.

**Page 20, Line 14 to Page 21, Line 2:** The value of $\alpha_O$ depends on how the transition from opaque to window region is defined. I am therefore not sure, how well the ECHAM6 results can be applied to the local-radiative-equilibrium model.

**Page 21, Lines 17-23:** Even with prescribed SST the troposphere may respond to counter the forcing. Therefore, it would be helpful to make additional columns in Table
I for the OLR and maybe also for the albedo. This is exactly the type of effect that could be more closely investigated with the methods suggested in the general comments.

**Page 22, Line 1-2:** I suggest weakening the first statement in the paragraph for three reasons. The statements, that this is possibly the simplest way to explain the effect, that the explanation is complete and that the explanation is physically correct, are too strong. In my opinion a more fitting description of the paper would be: "In this article, we try to explain a well-known phenomenon that is central to our general understanding of climate change – cooling of the middle atmosphere by $CO_2$ – in a simple but physically consistent way".

**Page 23, Line 8:** I suggest changing "the physical essence" in the last sentence, because the manuscript does not allow to exclude other potentially essential mechanisms leading to a stratospheric cooling.

---

## Referee Comment (RC2) · Anonymous Referee #2 · 12 May 2016

Review of Goessling and Bathainy: Why CO2 cools the middle atmosophere.

This paper is an interesting, scholarly, thorough and well-motivated piece of work, and I recommend that it be published subject to some modifications.

I have to say, though, that having worked in this field for some time, I did not feel it ultimately helped my intuition much beyond what I learn from a rather simpler model (below). But I appreciate that others (as evidenced by the comment already online) may so benefit. It could be that I am too stubborn with my simpler view, or too easily satisfied.

Main comments

1. My simple view: It is simply that the grey-body emission of the stratosphere 2e$\sigma$Tˆ4

(where e is the stratosphere emittance) is balanced by the heat source which is a combination of direct solar heating of the stratosphere (Sa) and absorption of upwelling infrared radiation from the surface and troposphere. In the CO2 case, where the upwelling radiation mostly originates from the cold upper troposphere, I would approximate this as $2e\sigma T^4 = Sa$, from which a cooling immediately follows when e increases. In the other ("CFC") limit, then clearly the absorption term can come to dominate, yielding a heating, or at least a greatly reduced cooling. I realise that this simple model is encapsulated in the authors' model, but the above seems a simpler expression of it, for those less familiar with radiative processes.

2. The GCM experiments with the removal of atmospheric solar absorption are very interesting, but the main result, the difference between REF and REF_ns is surprising to me – a cooling of just 1.8 K. If correct, this is noteworthy. But if we consider that about 70-80 W m-2 of solar radiation is absorbed by the atmosphere, one might guess that about 30% of this (the planetary albedo) would now be reflected back to space, as it is not now being absorbed. That constitutes a top of atmosphere forcing of maybe 20 W m-2, or 5 times the CO2 forcing. If my simple estimate is correct, how come such a small temperature change? It is possible that the increased absorption of UV/vis radiation by the surface/troposphere system when ozone absorption is removed, might compensate for the loss, but the stratospheric cooling would likely compensate for much of this (and I would guess much of the non-absorbed UV would instead be Rayleigh scattered to space). An alternative is that there may a mistake in the model set-up. It is not clear whether it is just the gaseous solar absorption that is set to zero, and the cloud liquid/ice absorption remains – if so, this would likely compensate strongly. It would be good to see how the planetary albedo changes between the two runs. Whatever the answer, this result needs some more discussion and perhaps there is some similar experiment in the literature that could be used to support this new result.

Other comments:

[Figure]

2:17-19 I do not quite see the "fails to explain"- the shortwave heating may be weaker than at the stratopause, but it remains substantial, otherwise the middle atmosphere would be much cooler (and would relax to a "polar night" radiatively-determined state).

2:34: "we show that this is not the case" – perhaps the authors could be clearer here. To my mind the CO2/CFC experiments show very clearly that the low upwelling flux at the tropopause is not very important in determining the CO2 cooling, to the extent that it is very insensitive to changes in that upwelling flux when surface temperature changes.

3:10 I agree that this simple model cannot explain the cooling, as the solar radiation is deposited at the surface. But my simple model above, has the solar radiation deposited within the stratosphere and does give a first-order cooling effect as (stratospheric) emittance increases.

4: Fig 1 caption – (a) perhaps say how normalised (I know the answer, but perhaps readers will not). (b) "vertical column" – it is clear in the appendix that this transmittance is simulated from a homogenous slab approach. I have no objection to this, as it is fine for the illustrative purpose used here, but I think the caption should make clear that this has been done – perhaps "assuming the troposphere and stratosphere to be homogeneous slabs".

4:6 "radiance" – since one is dealing with energetics, I feel this should be modelled as irradiance and not radiance, and indeed equations (1) and (5) seems a slightly odd mix of radiance and irradiance formulations, assuming the normal definition of the Stefan-Boltzmann constant. I'd slightly prefer to see a $\pi$ and a slant path formulation.

5:13 "insolation" – this is confusing because, at this stage, insolation is not represented. This is related to my irradiance/radiance comment above.

7: 5 "in an atmosphere where no solar radiation is absorbed".

8:7 "like the one" – of course, in the Earth's atmosphere the window is not perfectly

transparent, especially in moist atmospheres were the continuum absorption is strong but (because of the vapour pressure squared dependence of the continuum) the argument still holds as most of this absorption/emission is in the lower troposphere.

8:10 Note typos ??-??

10:4 I find "skin temperature" a strange name here, as this is also used for the topmost layer of the ocean. Perhaps another name could be used?

11:1 I would say "decadal to centennial" rather than "multi-centennial".

Section 4.2: I am always a bit suspicious about such analogies and don't really think they help the argument much more than a direct appeal to the actual physical situation at hand. I personally would delete this whole section.

15:17 "indirect" – I didn't quite understand why the solar effect was labelled as "indirect" – it seems rather direct to me, and of first order importance.

17:1 I had a similar feeling to Section 4.2 – I felt that this section could be removed as it seems hard to come up with truly realistic values given the idealised form of the equations derived to this point, especially given Section 5.2. (Part of my thinking is that the paper would be more easily "digestible" if it was a little shorter, although I appreciate the thoroughness – perhaps this section could be moved to an appendix or supplementary information?)

18:20 I have quite a lot of comments on this section

- The earlier parts of the paper were very thorough in reviewing the prior literature, but this section was less good – many of the results could already be anticipated from the prior literature and they should be mentioned/compared explicitly

- It is assumed, but not said, that this model configuration has fixed climatologically specified ozone – otherwise it would also respond to changes in temperature and CFCs. Similarly, the stratospheric water vapour is very sensitive to tropopause temperature (see for example Joshi et al. 10.5194/acp-10-7161-2010) and so might dramatically change in the no-solar runs, where there is so much cooling.

- 20:7-15 I found this discussion hard to follow. The "virtually unchanged" is not surprising from earlier calculations of the impact of CFC changes that the authors refer to, and results from a closer balance between increased absorption of upwelling radiation and increased emission. The "solar effect" is quite wavelength dependent, and so the second sentence needs to be clarified. But it seems to me that there is some expectation by the authors that the 2xCO2 and 15xCFC experiments, because they yield similar surface temperature change, should somehow be expected to yield similar stratospheric temperatures. But since these two gases are in very different regimes (strong and weak) at current tropospheric concentrations, I don't think such an equivalence should be anticipated –small changes in CFCs can have an equivalent effect to large changes in CO2 for surface temperature, but the situation is quite different in the stratosphere, when CO2 can more effectively cool to space from its band centre.

- At 21:20-23 there is not so much surprise that the stratospheric temperatures are not so sensitive to surface temperature change – this is shown, for example, in the figures in Forster et al. (1997). I am not sure that this is surprising (the authors say it is "not obvious"), partly because the change in upwelling radiation at the tropopause in the CO2 bands is rather small after a climate warming (most of the extra upwelling radiation will be at wavelengths where CO2 absorbs little). And so the following discussion on the possible role of water vapour feedback seems very speculative.

---

## Author Comment (AC1) · 1 Jul 2016

COMMENT: A study of this sort is sorely needed. As the authors identify in their introduction, the literature is lacking a clear explanation of why CO2 increases cool the middle atmosphere. Most explanations are either too simplistic to be accurate, or arise from complex model calculations so their physical interpretation is unclear. The manuscript is very clear and well-organised. I also appreciated the 'building' analogy for understanding the results. I have a few comments and suggestions.

REPLY: We are grateful for this comment which confirms us in our concern to clarify the reasons of CO2-induced stratospheric cooling. We also thank A. Ferraro for the constructive comments that we will address below.

COMMENT: P2, L9-10: " As ozone concentrations are expected to recover in future, it

seems likely that CO2 concentrations will be of growing importance also in the lower stratosphere." I find Cordero and Forster (2006) an unusual reference here as they don't seem to present model results for the 21st Century, during which ozone concentrations are expected to recover. Model results for this period have been presented by, e.g. Stolarski et al (2008) and Ferraro et al (2015).

REPLY: We thank A. Ferraro for pointing out these very interesting studies of which we were not aware. They are indeed more suitable to be cited. We will do this in the revised manuscript.

COMMENT: P5, L8 and L10: You seem to switch notation for optical depth from delta to tau here.

REPLY: We will use only one letter in the revised manuscript.

COMMENT: P10, L3: I am a little confused by your reference to "skin temperature" here. Equation 31 is for TOA temperature. I interpret skin temperature to mean the temperature of the surface.

REPLY: We have adopted the usage of "skin temperature" as a synonym for "TOA temperature" in our model, as for example in Pierrehumbert (2010; e.g., chapter 3.6). However, to avoid ambiguity, we will use "TOA temperature" consistently in the revised manuscript.

COMMENT: Figure 9: I find the text on the legend and axes labels a bit small here. Perhaps you might consider increasing the font size?

REPLY: We will increase the font size in the revised manuscript.

REFERENCES:

Ferraro, A., Collins, M. and Lambert, F. (2015). A hiatus in the stratosphere?. Nature Climate Change, 5(6), pp.497-498, doi:10.1038/nclimate2624.

Stolarski, R., Douglass, A., Newman, P., Pawson, S. and Schoeberl, M. (2010). Relative Contribution of Greenhouse Gases and Ozone-Depleting Substances to Temperature Trends in the Stratosphere: A Chemistry–Climate Model Study. Journal of Climate, 23(1), pp.28-42, doi:10.1175/2009JCLI2955.1.

Pierrehumbert, R. T.: Principles of Planetary Climate, Cambridge University Press, 2010.

---

## Author Comment (AC2) · 2 Jul 2016

COMMENT: The study aims at giving an educational perspective of the stratospheric cooling under increased CO2 concentrations. The paper is somewhat unique in that it is neither a classical research paper that puts forward new results or a new piece of theory, nor a review paper that purely synthetizes past findings. Instead the paper uses a simple model to illustrate different effects that contributes to the stratospheric cooling under increased CO2 concentrations. The results obtained with this simple model are not exceedingly interesting for people who worked with such models before. However, as pointed out by the authors, a publication with such a simple model on the topic is lacking and therefore the approach to the problem can be considered novel. The study will thus certainly be very useful to scientists that are interested in the topic but not particularly familiar with radiative transfer, especially since the manuscript is

well written and easy to follow. The study, however, still lacks a detailed discussion of how a convectively dominated troposphere affects the stratospheric cooling under increased CO2 concentrations. Without such a discussion, it is hard to evaluate in how far the effects described with the local-radiative-equilibrium model are responsible for the stratospheric cooling in comprehensive models. Therefore, I recommend publication after revising the manuscript to this end.

REPLY: We thank M. Popp for these very constructive comments. We are happy that the intended purpose of our paper seems to be clear. We will comment on the role of convection below.

COMMENT: The study makes use of a radiative-equilibrium model for the entirety of the atmosphere. These model are usually rather referred to as "local-radiative-equilibrium models" than as "radiative-equilibrium models", to indicate that the atmosphere is in radiative equilibrium everywhere. Such models give good results for optically thin regions of the atmosphere. However, the quality of results is difficult to assess in regions that are strongly influenced by convection such as Earth's troposphere. Changes in Earth's troposphere will most certainly also affect the response of the stratosphere to an increase in CO2. Therefore, the manuscript would be strengthened, if the local radiative-equilibrium model was discussed in more detail in the context of present day Earth, and if the role of the troposphere in cooling the stratosphere was quantitatively addressed. Here are a few suggestions on how this could be done and for some potentially interesting aspects to discuss: An easy way of incorporating the troposphere into the gray version of the model is to define the tropopause at the level $h = h_{TP}$. Applying this definition to equation (8) of the manuscript yields (....) These equations can now serve as a starting point for the investigation of the influence of the upwelling longwave-radiation from the troposphere on the stratospheric response to an increase in CO2 concentrations. This formulation decouples the stratosphere from the surface temperature and allows for different assumptions on how the changes at the surface impact the upwelling longwave radiation at the tropopause. A potentially interesting

case to study would for example be the one of constant tropopause temperature, which is a common feature in climate-change simulations. Another interesting consideration would be, how the deepening of the troposphere and hence the increase of hTP influences the results. The authors could also apply some of the values they obtained from the simulations with ECHAM6 to .... ect.. A discussion of the troposphere would furthermore allow to discuss changes in albedo A through changes in clouds. With A included, the TOA radiative balance simply writes as (....) Changes in A would also lead to a change in the steady-state stratospheric temperature even in the gray case without window.

REPLY: The role of the assumption of local radiative equilibrium in our model is indeed an important point which we will make clearer in the revised manuscript. The reviewer wonders how adjustments in the structure of the troposphere during the time of surface warming, in particular a change in the tropopause height, affect our results. Interestingly, these adjustments of the troposphere have no substantial impact on the stratospheric temperature profile, and thus also the contribution of effects. This fact manifests itself in Fig. 9: Above a height of 20 km, there is no difference between the simulations with fixed surface temperature, and the simulations where the surface-troposphere system has adjusted. Hence, the temperature profile of the middle and upper atmosphere only depends on the local chemical composition, and not on the temperature profile in the troposphere. Above the tropopause, the only difference between a simulation with a surface in equilibrium and a fixed SST simulation is a small additional upwelling long-wave flux of approx. 3.5 W/m2. This additional flux however has no impact on the stratosphere because it goes through the atmospheric window. All absorption has happened at lower altitudes. Stratospheric temperature, and the absorption of long-wave radiation thus do not differ between the simulations. This result is very generic and not specific to our simulations. To convince ourselves and the reviewer, we also looked into the models used for CMIP5. They show the same behaviour: When CO2 is quadrupled instantly (simulation abrupt4xCO2), the tropospheric temperature profile remains very close to pre-industrial conditions during the

first model years due to the ocean's inertia, but the profile above the tropopause shows a large cooling. Thereafter, the surface-troposphere system slowly warms, while the temperature profile above does not change anymore over time. This result is also in line with previous studies. Most importantly, Forster et al., 1997 use a radiative-convective model and show that the radiative forcing by $CO_2$ depends on the definition of the tropopause. While this affects the response of the surface-troposphere system, the temperature profile above is not affected by the definition of the tropopause (see their Fig. 8a). A similar argumentation applies to changes in surface albedo: The latter would affect the surface temperature directly and lead to an adjustment of the troposphere, but not to any adjustment of the middle atmosphere as the chemical composition there remains fixed. As changes in the tropospheric profile over time obviously do not matter for the middle atmosphere, biases in the tropospheric profile will not affect our results either. We agree that our assumption of radiative equilibrium is wrong in the sense that it yields an unrealistic temperature profile in the lower atmosphere. However, the aim of our analysis is to explain the response of the middle atmosphere. In these heights, the assumption of radiative equilibrium is much better justified, and our model is thus suitable for its purpose even though it does not have a troposphere. It is an interesting point put forward by the reviewer that one could therefore apply our model to the middle atmosphere directly, with the upwelling long-wave flux at the tropopause as a lower boundary condition. We are certainly going to discuss this in the revised manuscript and take up the modified equations suggested by the reviewer. As reviewer 2 pointed out that the insensitivity of the middle atmosphere to climate change in the surface-troposphere system is not a surprising result and should not be presented as such, we will not elaborate on this result too much in the revised manuscript and see little benefit from an extended derivation of sensitivities to changes in tropopause height or temperature due to the arguments outlined above. As our model is very simple in many ways, it does not allow a quantitative separation of the two reasons we discuss (as well as the transient part of the blocking effect). The lack of a troposphere is not the only reason for this as there are many other simplifications in the model. Applying the

simple model only to the stratosphere will therefore not suffice to separate the effects quantitatively. This is the reason we apply the general circulation model, which shows that the transient effect is indeed practically absent, justifying the value of our model. We will make this clearer in the revised manuscript.

COMMENT: Page 2, Line 27: It would be helpful for interested readers, if the relevant Chapter in Goody and Yung (1989) would be indicated; Page 4, equation (1): It may be worth mentioning, that equation (1) is obtained by making an assumption on mean angle of the thermal radiation against the vertical; Page 5, Line 15: Equation (3) is the spectrally integrated, gray-absorption-case of Schwarzschild's equation.

REPLY: We will mention all three points in the revised manuscript.

COMMENT: Page 6, Lines 14-17: The last sentence of this paragraph suggests that the local radiative equilibrium is a decent approximation for present-day Earth. This is generally not the case. Despite sharp gradients existing close to the surface on Earth, the discontinuity and the lapse rate of the local-radiative-equilibrium model are by far too large. I suggest mentioning this.

REPLY: We intended to say exactly this – we will make it clearer in the revised manuscript.

COMMENT: Page 10: I suggest to introduce equation (35) just after equation (31), because equation (35) was quite important for me to understand the following discussion.

REPLY: Our intention with the current structure was to first explain the mechanism of the blocking effect, and then how it changes over time in the model. From our own perspective, Eq. (35) is not so important to understand the text before, but we can easily move the paragraph on page 13, lines 5-13 to the beginning of the section (starting on page 10, line 16). We hope that this will improve the clarity of our paper.

COMMENT: Page 10, Line 17 to Page 11, Line 1: This is one of the situations where a comparison between the local-radiative-equilibrium model and Earth is awkward. The

stratosphere and the tropopause do not adjust at the same time-scale.

REPLY: This separation of time scales is exactly what we analyse here. Compared to the surface temperature that changes over decades, the atmospheric radiation field responds very quickly to composition changes. Indeed, the limitation of the model is that there is no tropopause, and we will make this clear again at this point. However, the separation of time scales can still be discussed in the model by keeping the surface temperature fixed as we do in this section.

COMMENT: Page 11, Line 26: It may be worth mentioning that this is entirely a consequence of the outgoing longwave radiation (OLR) having to balance S, which is constant in steady state. As a consequence, even in this simple model a solar and a greenhouse forcing act differently: An increase in S would force the OLR to increase as well and, as a consequence, the Skin-temperature would have to increase. In contrast, with this simple model the OLR does not change when $CO_2$ is applied.

REPLY: We fully agree with this comment and will take it up in the revised version of the manuscript.

COMMENT: Page 12, Line 11: The definition of "blocking effect" may be a bit confusing, because the authors already discussed the "increased blocking" in the window-less case. Speaking of "transient blocking effect" and "equilibrium blocking effect" may be a possible way of resolving the possible confusion. Note, that it is also arguable that in this case the "blocking effect" applies to present-day Earth, because the surface temperature on Earth has not equilibrated yet.

REPLY: We understand that the naming of effects can be confusing, and had in fact changed the wording also before submission of the manuscript. We decided to name the effects based on their physical mechanism (blocking effect and solar effect). We will make clearer where we address the equilibrium blocking effect, and where the transient contribution. The parallel to the present-day Earth is indeed noteworthy, though it should also be pointed out that we change $CO_2$ abruptly in the model, while it accumulates slowly in reality. Moreover, there is hardly any transient contribution of the blocking effect in complex models (see discussion of main comments above).

COMMENT: Page 17, Line 8: 6,5 K / 1000 m instead of 6,5 K.

REPLY: We will correct this mistake in the revised manuscript.

COMMENT: Page 17, Lines 7-9: I am not sure whether it can be stated that there is an upper limit of the lapse rate of 6,5 K / 1000m. If, for example, the global-mean surface-temperature would change, the lapse rate may change as well. The fact, that the global-mean lapse rate is 6,5 K / 1000 m, does not necessarily mean that this value is a an upper global-mean limit.

REPLY: We agree with this comment, and will rephrase this to something like "Convection acts to reduce the lapse rate considerably, to approx. 6.5K/km in the current climate".

COMMENT: Page 17, Lines 11-27: I suggest replacing this paragraph, by a discussion along the lines made in the general comments. I doubt that much meaningful can be learnt from applying typical values for present-day Earth to the local-radiative-equilibrium model without decoupling the surface-temperature from the TOA-temperature to some degree.

REPLY: We understand that this discussion can appear too speculative. What we mean to discuss here is the fact that the surface temperature response in the simple model is too large relative to the stratospheric response compared to more realistic models. This might be assumed to just result from the negligence of convection. However, we argue that the vertically inhomogeneous distribution of water vapour must be an aspect that cannot be neglected. The atmospheric window above the tropopause is much larger than at the surface. We use the model to demonstrate that this makes a huge difference for the ratio of the temperature responses. We think that this result is meaningful despite the negligence of convection because convection does not change

the fact that water vapour is most abundant near the ground. We will certainly rephrase this section to make this clearer, and will include the discussion of the model as applied to the middle atmosphere only. To this end, we will remove Sect. 5.1 and discuss the role of water vapour in a new Sect. 5.2 after the analysis of the ECHAM6 simulations (as reviewer 2 suggests).

COMMENT: Page 18, Line 19: I suggest replacing "confirmed" by "supported". Model results are no observations.

REPLY: We agree and will change this in the revised manuscript.

COMMENT: Page 20, Line 14 to Page 21, Line 2: The value of $\alpha\_O$ depends on how the transition from opaque to window region is defined. I am therefore not sure, how well the ECHAM6 results can be applied to the local-radiative-equilibrium model.

REPLY: Indeed, what we mean to say here is that the distinction of parameters alpha and beta, as well as their separate variation in the simple model, is an idealisation. This approach would not work in the complex model, where there are no such parameters, and the comparison between the models can only be qualitative in the sense that we compare the effect of CFCs and CO2. It is not possible to identify a unique realistic value for alpha, as the reviewer says. We therefore have the impression that we fully agree with the reviewer on this point. We do not see though how this would constitute a problem here because we do not fit the model parameters to ECHAM6.

COMMENT: Page 21, Lines 17-23: Even with prescribed SST the troposphere may respond to counter the forcing. Therefore, it would be helpful to make additional columns in Table I for the OLR and maybe also for the albedo. This is exactly the type of effect that could be more closely investigated with the methods suggested in the general comments.

REPLY: We agree that it will help to add such quantities to Tab. 1 and will do so in the revised manuscript. However, we are not certain how the first sentence of this comment

is meant. If the troposphere acted to "counter the [radiative] forcing", this would imply no radiative imbalance, meaning that surface temperatures would not change even if surface temperatures were dynamic – which is obviously not the case (as the climate sensitivity is non-zero). Moreover, as can be seen in Tab. 1, the global mean surface air temperature changes very little when SSTs are fixed (0.28K for CO2, 0.15K for CFCs). As we explained above (and as can be seen in Fig. 9), changes in the troposphere have a negligible effect on the stratosphere, which is even more true for such small temperature changes near the surface.

COMMENT: Page 22, Line 1-2: I suggest weakening the first statement in the paragraph for three reasons. The statements, that this is possibly the simplest way to explain the effect, that the explanation is complete and that the explanation is physically correct, are too strong. In my opinion a more fitting description of the paper would be: "In this article, we try to explain a well-known phenomenon that is central to our general understanding of climate change – cooling of the middle atmosphere by CO2 – in a simple but physically consistent way".

REPLY: We will take up this suggestion in the revised manuscript.

COMMENT: Page 23, Line 8: I suggest changing "the physical essence" in the last sentence, because the manuscript does not allow to exclude other potentially essential mechanisms leading to a stratospheric cooling.

REPLY: We chose this wording based on previous literature explaining CO2-induced stratospheric cooling. It is true that we do not show in our manuscript that these are the only relevant mechanisms. We only formalise the heuristic arguments from previous studies and textbooks. However, we are not aware of any literature indicating that other effects are of similar importance (or will be in the future), and therefore see no reason against the statement that our article has "the potential to convey the physical essence". To leave room for other mechanisms we can replace "this important phenomenon" by a reference to the mechanisms instead of the whole phenomenon, e.g.

"The reconsideration of CO2-induced MA cooling as put forward here has a distinct educational element, with the potential to convey the physical essence of the involved mechanisms to a broader audience."

REFERENCES:

Forster, P. M. F., Freckleton, R. S., and Shine, K. P.: On aspects of the concept of radiative forcing, Clim. Dyn., 13, 547–560, doi:10.1007/s003820050182, 1997.

---

## Author Comment (AC3) · 2 Jul 2016

COMMENT: This paper is an interesting, scholarly, thorough and well-motivated piece of work, and I recommend that it be published subject to some modifications. I have to say, though, that having worked in this field for some time, I did not feel it ultimately helped my intuition much beyond what I learn from a rather simpler model (below). But I appreciate that others (as evidenced by the comment already online) may so benefit. It could be that I am too stubborn with my simpler view, or too easily satisfied.

REPLY: We thank the reviewer for the very helpful and constructive comments which will help us to improve the manuscript. We comment on the reviewer's simple model explanation below.

COMMENT: My simple view: It is simply that the grey-body emission of the strato-

[Figure]

sphere 2eT**4 (where e is the stratosphere emittance) is balanced by the heat source which is a combination of direct solar heating of the stratosphere (Sa) and absorption of upwelling infrared radiation from the surface and troposphere. In the CO2 case, where the upwelling radiation mostly originates from the cold upper troposphere, I would approximate this as 2eT**4=Sa, from which a cooling immediately follows when e increases. In the other ("CFC") limit, then clearly the absorption term can come to dominate, yielding a heating, or at least a greatly reduced cooling. I realise that this simple model is encapsulated in the authors' model, but the above seems a simpler expression of it, for those less familiar with radiative processes.

REPLY: We agree that the reviewer's explanation is adequate and very simple, and we will take up this very nice summary in our conclusions section. However, we see this only as an explanation that requires one to know the answer already, as compared to a model where this result is derived based on certain assumptions and boundary conditions. We therefore think that our more elaborate approach is justified. In particular, the reviewer's approach only yields an instantaneous radiative perturbation, but does not allow to calculate how the atmospheric profile would be altered in equilibrium when the fluxes and temperatures adjust. Moreover, the simple explanation alone would not allow to demonstrate the importance of the atmospheric window to explain CO2-induced cooling of the middle atmosphere (the grey model yields no cooling in equilibrium). To derive these mechanisms and the fact that CO2 and CFCs act differently, one needs to couple the radiative fluxes to the temperature and solve for the temperature profile. Our model thus connects well to the radiative two-layer model and its variants that are very popular in educational contexts. We therefore hope that our model (though more complicated than the reviewer's explanation) can also help non-experts to build a deeper understanding.

COMMENT: The GCM experiments with the removal of atmospheric solar absorption are very interesting, but the main result, the difference between REF and REF_ns is surprising to me – a cooling of just 1.8 K. If correct, this is noteworthy. But if we consider that about 70-80 W m-2 of solar radiation is absorbed by the atmosphere, one might guess that about 30% of this (the planetary albedo) would now be reflected back to space, as it is not now being absorbed. That constitutes a top of atmosphere forcing of maybe 20 W m-2, or 5 times the CO2 forcing. If my simple estimate is correct, how come such a small temperature change? It is possible that the increased absorption of UV/vis radiation by the surface/troposphere system when ozone absorption is removed, might compensate for the loss, but the stratospheric cooling would likely compensate for much of this (and I would guess much of the non-absorbed UV would instead be Rayleigh scattered to space). An alternative is that there may a mistake in the model set-up. It is not clear whether it is just the gaseous solar absorption that is set to zero, and the cloud liquid/ice absorption remains – if so, this would likely compensate strongly. It would be good to see how the planetary albedo changes between the two runs. Whatever the answer, this result needs some more discussion and perhaps there is some similar experiment in the literature that could be used to support this new result.

REPLY: We agree that, given the major interference with the atmospheric radiation budget, the small tropospheric temperature change is not at all obvious. As the reviewer suspects, a substantial energy source in the atmosphere is missing in the non-solar simulations; however, a large part of it is absorbed at the surface instead. We will explain the associated processes in more detail in the revised manuscript, and add some quantitative information to support our arguments. (Note that the exact numbers of the global mean temperatures in Table 1 will slightly change in the revised manuscript because we repeated the simulations with a newer version of Echam6 in order to obtain more model diagnostics for our answer).

What we did in the experiments labelled as "ns" is indeed to switch off the solar absorption by all gases, but not cloud droplets or ice. This leads to a reduction of short-wave absorption in the atmosphere from 75 W/m2 to only 13 W/m2, and nothing is absorbed anymore above the tropopause, as intended. We are not aware of identical experi-

ments in previous studies. To this extent, our result is probably indeed new, although the response to switching off atmospheric solar absorption (by gases) is not meant to be the focus of our study, and we thus would not call this the "main result".

Our results can be understood in a similar way as stratospheric ozone removal experiments, for example Ramaswamy et al. (1992), Hansen et al. (1997), Forster and Shine (1997) and Stuber et al. (2001). These and other studies agree that the instantaneous radiative forcing of ozone removal is positive as more solar radiation reaches the surface. In such ozone removal experiments, there is also an instantaneous longwave effect due to the nature of ozone as a greenhouse gas, which is not active in our experiments where we only switch off short-wave absorption. The instantaneous long-wave effect after ozone removal in the other studies is negative, but found to be weaker than the short-wave effect (Ramaswamy et al., 1992; Forster and Shine, 1997). Hence, the instantaneous radiative forcing is positive in these studies as well as our own. However, the lack of solar absorption by ozone leads to a very strong cooling of the stratosphere, which reduces the downwelling long-wave radiation from the atmosphere to the surface (due to the greenhouse effect of all present greenhouse gases, not only ozone). After this temperature adjustment, the forcing has reversed its sign, i.e. becomes negative. This explains the cooling despite the increase in solar absorption at the surface.

Another difference between these studies and ours is that we also switch off the absorption of solar radiation by other gases than ozone. However, a similar argument as above holds for the solar absorption by other well-mixed gases: more sunlight reaches the ground (+42 W/m2), but this is counteracted by a decrease in downwelling long-wave radiation and an increase in the turbulent fluxes.

Finally, a large fraction of absorbed solar radiation in the atmosphere is due to water vapour (as our Fig. 1 c indicates). This gas mostly occurs in the lower levels of the troposphere (difference between Fig. 1 b and c) where the temperature is already close to the surface air temperature, and where turbulent fluxes redistribute perturbations vertically. The shift of the absorbed solar absorption from the lower troposphere to the

surface has therefore only little effect on the surface air temperature.

In total, the effect of the reduced downwelling long-wave effect from the strongly cooled stratosphere slightly dominates the sign of the surface air temperature response in our simulations. The total flux of shortwave radiation absorbed by the real present-day atmosphere that the reviewer mentions should therefore not be expected to have a similarly large effect compared to a radiative forcing by $CO_2$ of the same magnitude, because a radiative forcing of well-mixed greenhouse gases is defined at the tropopause; the radiative forcing concept appears to be misleading when a vertical redistribution of heating across the tropopause is involved.

The effect of increased scattering that the reviewer mentions also contributes to cool the surface, though perhaps not as much as the reviewer speculates: The planetary albedo increases from 29% to 33% after switching off the absorption by gases, and the outgoing short-wave radiation at the top of the atmosphere increases from 98.8 W/m2 to 112.6 W/m2.

We understand that an explanation of the experiment and these effects will improve the understanding of our manuscript, and we will add an according paragraph and extend Table 1 as suggested.

COMMENT: 2:17-19 I do not quite see the "fails to explain"- the shortwave heating may be weaker than at the stratopause, but it remains substantial, otherwise the middle atmosphere would be much cooler (and would relax to a "polar night" radiatively-determined state).

REPLY: We meant to say that the argument fails to explain why the $CO_2$-induced cooling does not become weaker beyond the stratopause despite the weakening shortwave heating. We do agree that this reasoning is rather subtle, in particular because observed heating rates beyond the stratopause are quite uncertain. We will rephrase this or leave the sentence out.

COMMENT: 2:34: "we show that this is not the case" – perhaps the authors could be clearer here. To my mind the CO2/CFC experiments show very clearly that the low upwelling flux at the tropopause is not very important in determining the CO2 cooling, to the extent that it is very insensitive to changes in that upwelling flux when surface temperature changes.

REPLY: What we mean here is that the existence of a temperature minimum in the profile is not required to explain CO2-induced cooling. Our model does not have a tropopause but still captures MA cooling, therefore we argue that the model proves this point. We will make this clearer in the revised manuscript. We fully agree that the tropospheric adjustment to radiative forcing does not affect the middle atmosphere much, but this is not the point we intended to make here.

COMMENT: 3:10 I agree that this simple model cannot explain the cooling, as the solar radiation is deposited at the surface. But my simple model above, has the solar radiation deposited within the stratosphere and does give a first-order cooling effect as (stratospheric) emittance increases.

REPLY: We agree, but argue that it's important to note first that the very popular grey-atmosphere model (transparent to solar radiation) cannot exlain the cooling. The model put forward by the reviewer is already slightly more complex by accounting for the absorption of solar radiation within the atmosphere. Furthermore, we argue that the reviewer's heuristic model, though correct, is not an energy balance model of the full atmosphere and thus does not yield equilibrium temperature changes, only instantaneous local responses to radiative perturbations.

COMMENT: 4: Fig 1 caption – (a) perhaps say how normalised (I know the answer, but perhaps readers will not). (b) "vertical column" – it is clear in the appendix that this transmittance is simulated from a homogenous slab approach. I have no objection to this, as it is fine for the illustrative purpose used here, but I think the caption should make clear that this has been done – perhaps "assuming the troposphere and

stratosphere to be homogeneous slabs".

REPLY: We will take up these suggestions in the revised manuscript.

COMMENT: 4:6 "radiance" – since one is dealing with energetics, I feel this should be modelled as irradiance and not radiance, and indeed equations (1) and (5) seems a slightly odd mix of radiance and irradiance formulations, assuming the normal definition of the Stefan-Boltzmann constant. I'd slightly prefer to see a and a slant path formulation.

REPLY: We agree that we should distinguish radiances and irradiances more clearly here and will correct this in the revised manuscript. We will introduce one more step in the derivation to first go from an arbitrary direction of a radiance (in W/m2sr) to the vertical direction (introducing the geometric factor mu mentioned later), and will then integrate over the half sphere to obtain irradiances in W/m2. The factor pi would then indeed occur before the source term J, and this product yields sigma T**4.

COMMENT: 5:13 "insolation" – this is confusing because, at this stage, insolation is not represented. This is related to my irradiance/radiance comment above.

REPLY: With insolation we do not mean solar insolation here, but the source term J which only describes the emission of other parts of the atmosphere according to Planck's law. We will make this clearer.

COMMENT: 7: 5 "in an atmosphere where no solar radiation is absorbed".

REPLY: We agree that this formulation is clearer and we will take it up.

COMMENT: 8:7 "like the one" – of course, in the Earth's atmosphere the window is not perfectly transparent, especially in moist atmospheres were the continuum absorption is strong but (because of the vapour pressure squared dependence of the continuum) the argument still holds as most of this absorption/emission is in the lower troposphere.

REPLY: We will make clearer that the notion of "a window" where all radiation is transmitted is idealised.

COMMENT: 8:10 Note typos ??-??

REPLY: We will correct this mistake in the revised manuscript.

COMMENT: 10:4 I find "skin temperature" a strange name here, as this is also used for the topmost layer of the ocean. Perhaps another name could be used?

REPLY: We have adopted the usage of "skin temperature" as a synonym for "TOA temperature" in our model, as for example in Pierrehumbert (2010; e.g., chapter 3.6). However, to avoid ambiguity, we will use "TOA temperature" consistently in the revised manuscript.

COMMENT: 11:1 I would say "decadal to centennial" rather than "multi-centennial".

REPLY: We will change this accordingly in the revised manuscript.

COMMENT: Section 4.2: I am always a bit suspicious about such analogies and don't really think they help the argument much more than a direct appeal to the actual physical situation at hand. I personally would delete this whole section.

REPLY: We understand that among experts such analogies may sometimes cause more suspicion than illumination. However, we are convinced that this section can be useful for readers who are no experts in atmospheric radiation but who want to understand and remember the mechanism of the blocking effect. These readers are an important target group of our paper. The analogy will also be useful for readers who do not want to go through the many equations, but still look for a quick explanation. We feel confirmed in this view by the Short Comment posted by A. Ferraro in the open discussion. We therefore prefer to keep the analogy, but we realise that it may seem a bit isolated from the more technical rest of the paper. Hence, we are going to move the building analogy to an appendix.

COMMENT: 15:17 "indirect" – I didn't quite understand why the solar effect was labelled
as "indirect" – it seems rather direct to me, and of first order importance.

REPLY: If we termed this effect just a "solar effect", it could easily be confused as a result of reduced shortwave heating. In our view the term "indirect" helps to prevent that confusion, and we also do not think that "indirect" would be mistaken as meaning "weak". We would therefore like to keep the terminology, but will add a clarifying remark.

COMMENT: 17:1 I had a similar feeling to Section 4.2 – I felt that this section could be removed as it seems hard to come up with truly realistic values given the idealised form of the equations derived to this point, especially given Section 5.2. (Part of my thinking is that the paper would be more easily "digestible" if it was a little shorter, although I appreciate the thoroughness – perhaps this section could be moved to an appendix or supplementary information?)

REPLY: Given the criticism of both reviewers, we are going to remove this section, and discuss the most important differences between the simple model and ECHAM6 in a new section after the presentation of the simulations.

COMMENT: 18:20 I have quite a lot of comments on this section. The earlier parts of the paper were very thorough in reviewing the prior literature, but this section was less good – many of the results could already be anticipated from the prior literature and they should be mentioned/compared explicitly

REPLY: We understand that we can refer to previous literature at this point, some of which we already cite earlier in the manuscript. As discussed in our response to the reviewer's main comment 2, we will discuss the simulations without solar absorption in more detail, and refer to the relevant papers mentioned above. We can also refer more explicitly to previous papers that investigated the effect of $CO_2$ and CFCs on the temperature profile.

COMMENT: It is assumed, but not said, that this model configuration has fixed climato-
logically specified ozone – otherwise it would also respond to changes in temperature and CFCs. Similarly, the stratospheric water vapour is very sensitive to tropopause temperature (see for example Joshi et al. 10.5194/acp-10-7161-2010) and so might dramatically change in the no-solar runs, where there is so much cooling.

REPLY: Indeed, the ozone concentrations are fixed to a climatology in the model. We will mention this in the revised manuscript. Regarding the effect of stratospheric water vapour, indeed the stratospheric humidity in the non-solar simulations decreases due to the very low temperatures (though the changes are tiny compared to the tropospheric changes), and this could have a contribution to the reduced long-wave emission and the cooling at the Earth's surface. We will mention this effect together with the others discussed above (though as far as we can see, it is not possible to clearly separate the effects in the model output).

COMMENT: 20:7-15 I found this discussion hard to follow. The "virtually unchanged" is not surprising from earlier calculations of the impact of CFC changes that the authors refer to, and results from a closer balance between increased absorption of upwelling radiation and increased emission. The "solar effect" is quite wavelength dependent, and so the second sentence needs to be clarified. But it seems to me that there is some expectation by the authors that the 2xCO2 and 15xCFC experiments, because they yield similar surface temperature change, should somehow be expected to yield similar stratospheric temperatures. But since these two gases are in very different regimes (strong and weak) at current tropospheric concentrations, I don't think such an equivalence should be anticipated –small changes in CFCs can have an equivalent effect to large changes in CO2 for surface temperature, but the situation is quite different in the stratosphere, when CO2 can more effectively cool to space from its band centre.

REPLY: We fully agree with the reviewer's explanation, and did not intend to say that we anticipated any other result. It is not the aim of our paper to present the differences in CO2 and CFCs as any new or surprising result, but to formalise the explanation in a

simple but comprehensive way. Our main point here is that the simple textbook expla-
nation for stratospheric cooling, where solar and longwave radiation are distinguished
but no further distinction in spectral characteristics is made, does not work for CFCs.
Apparently the according paragraph does not make this clear enough. It will benefit
from a rephrasing and from adding references to previous studies as suggested by the
reviewer above.

COMMENT: At 21:20-23 there is not so much surprise that the stratospheric tempera-
tures are not so sensitive to surface temperature change – this is shown, for example,
in the figures in Forster et al. (1997). I am not sure that this is surprising (the au-
thors say it is "not obvious"), partly because the change in upwelling radiation at the
tropopause in the CO2 bands is rather small after a climate warming (most of the extra
upwelling radiation will be at wavelengths where CO2 absorbs little). And so the follow-
ing discussion on the possible role of water vapour feedback seems very speculative.

REPLY: What we intend to say here is that the result is different from the window-grey
model which predicts a transient effect. We will rephrase this section to clarify that,
taking the cited literature into account, this is not overly surprising. Yet we are not aware
of any publication that focusses on this interesting point of a lacking transient effect, and
the mechanisms involved, in comprehensive models. The lack of a transient effect can
be clearly seen in Fig. 8a of Forster et al. (1997) like the reviewer says, but it is not
explained or analysed further. We are therefore convinced that this result is noteworthy.
We think that it is safe to argue that the water-vapour feedback must contribute to
the fact that the stratospheric response to surface warming is small compared to the
window-grey model, because it allows for a large surface temperature change despite
a relatively small long-wave flux change at the tropopause. The reason we mention this
effect is that the window-grey model cannot be tuned in a way to show realistic changes
in the temperatures and a negligible transient effect at the same time. It still appears to
us that the mere presence of an atmospheric window (the effect the reviewer refers to)
cannot explain why there should be no transient effect; after all, the window-grey model

does have a transient MA adjustment despite an atmospheric window. Nevertheless we agree that this aspect of the discussion is not a proven result but a (well-motivated) speculation. We will make this point clearer in the revised manuscript and keep the speculative part to a minimum.

REFERENCES:

Forster, P. M., and Shine, K. P.: Radiative forcing and temperature trends from stratospheric ozone changes, J. Geophys. Res., 102, 10841-10855, 1997.

Forster, P. M. F., Freckleton, R. S., and Shine, K. P.: On aspects of the concept of radiative forcing, Clim. Dyn., 13, 547–560, doi:10.1007/s003820050182, 1997.

Hansen, J., Sato, M., and Ruedy, R.: Radiative forcing and climate response, J. Geophys. Res., 102, 6831-6864, 1997.

Pierrehumbert, R. T.: Principles of Planetary Climate, Cambridge University Press, 2010.

Ramaswamy, V., Schwarzkopf, M. D., and Shine, K. P.: Radiative forcing of climate from halocarbon-induced global stratospheric ozone loss, Nature, 355, 810-812, 1992.

Stuber, N., Sausen, R., and Ponater, M.: Stratosphere adjusted radiative forcing calculations in a comprehensive climate model, Theor. Appl. Climatol., 68, 125-135, 2001.

---

## Editor Comment (EC1) · M. Crucifix (Editor) · 4 Jul 2016

I would like to thank everyone for the constructive discussion along this article. As noted by the reviewers and acknowledged by the authors, the ambition with this article is mainly to bring an educative prospective on a process that is well understood by specialists. Such objective may be a little bit unusual for Earth System dynamics, but the approach of the authors is nevertheless original and the case that the article brings added value is convincing. It should therefore be published once the different remarks have been addressed by the authors.

---

## Author Response (AR1)

We refer to the point-to-point responses provided in the public discussion.